# Dynamic regulation of mitotic ubiquitin ligase APC/C by coordinated Plx1 kinase and PP2A phosphatase action on a flexible Apc1 loop

Kazuyuki Fujimitsu & Hiroyuki Yamano*

## Abstract

The anaphase-promoting complex/cyclosome (APC/C), a multi-subunit ubiquitin ligase essential for cell cycle control, is regulated by reversible phosphorylation. APC/C phosphorylation by cyclin-dependent kinase 1 (Cdk1) promotes Cdc20 co-activator loading in mitosis to form active APC/C-Cdc20. However, detailed phospho-regulation of APC/C dynamics through other kinases and phosphatases is still poorly understood. Here, we show that an interplay between polo-like kinase (Plx1) and PP2A-B56 phosphatase on a flexible loop domain of the subunit Apc1 (Apc1-loop$^{500}$) controls APC/C activity and mitotic progression. Plx1 directly binds to the Apc1-loop$^{500}$ in a phosphorylation-dependent manner and promotes the formation of APC/C-Cdc20 via Apc3 phosphorylation. Upon phosphorylation of loop residue T532, PP2A-B56 is recruited to the Apc1-loop$^{500}$ and differentially promotes dissociation of Plx1 and PP2A-B56 through dephosphorylation of Plx1-binding sites. Stable Plx1 binding, which prevents PP2A-B56 recruitment, prematurely activates the APC/C and delays APC/C dephosphorylation during mitotic exit. Furthermore, the phosphorylation status of the Apc1-loop$^{500}$ is controlled by distant Apc3-loop phosphorylation. Our study suggests that phosphorylation-dependent feedback regulation through flexible loop domains within a macromolecular complex coordinates the activity and dynamics of the APC/C during the cell cycle.

**Keywords** APC/C; Cdc20; loop domain; polo-like kinase; PP2A
**Subject Categories** Cell Cycle; Post-translational Modifications & Proteolysis
**The EMBO Journal (2021) 40: e107516**

## Introduction

Cells use a highly conserved ordered mechanism, cell cycle control, to ensure that complete genomes are accurately duplicated and transmitted to daughter cells. Reversible phosphorylation is crucial for regulation of the activities of proteins involved in the cell cycle. Cyclin-dependent kinases (Cdks) are arguably the most prominent kinases in cell cycle progression. Cdks can trigger cascades involving kinases and phosphatases, which leads to activation of functional proteins or protein complexes involved in the execution of specific cell cycle processes (Morgan, 2007; Uhlmann *et al*, 2011; Saurin, 2018; Holder *et al*, 2019). In mitosis, Cdk1-cyclin B complex phosphorylates and activates the anaphase-promoting complex/cyclosome (APC/C) E3 ubiquitin ligase (Morgan, 2007; Pines, 2011; Primorac & Musacchio, 2013; Alfieri *et al*, 2017). The activated APC/C triggers the transition from metaphase to anaphase by targeting securin and cyclin B for proteasome-dependent degradation, which results in chromosome separation and inactivation of Cdk1, respectively. Upon inactivation of Cdk1, phosphatases are activated and the APC/C is dephosphorylated during mitotic exit. Thus, the phosphorylation states of the APC/C dynamically change in a manner coordinated with the progression of mitosis. However, it is still poorly understood how phosphorylation of APC/C is precisely regulated during mitosis, a key element of the working of the APC/C system.

The APC/C is a multi-subunit complex composed of 14 subunits and requires one of the Cdc20 family of APC/C co-activators, such as Cdc20 or Cdh1, for catalytic activation of the E3 ubiquitin ligase and substrate recognition of the APC/C (Pines, 2011; Primorac & Musacchio, 2013; Alfieri *et al*, 2017). The engagement of co-activators is a key step in the regulation of the APC/C system. Intriguingly, phosphorylation of APC/C is only crucial for the engagement of a mitotic co-activator Cdc20, not for an interphase co-activator Cdh1. Recent studies have revealed that several flexible loop domains of APC/C subunits play important roles in phospho-regulation of APC/C-Cdc20 complex formation (Fujimitsu *et al*, 2016; Qiao *et al*, 2016; Zhang *et al*, 2016a). Flexible loop domains of Apc1 and Apc3 (Apc1-loop$^{300}$, 294–399; Apc3-loop, 182–451 in *Xenopus*) play pivotal roles in the Cdk1-driven activation of APC/C. The Apc1-loop$^{300}$ is an auto-inhibitory domain blocking Cdc20 binding. When Apc1-loop$^{300}$ is unphosphorylated, Apc1-loop$^{300}$ resides on a site on Apc8B and blocks access of the C-box, an essential binding motif in Cdc20. Upon phosphorylation, the inhibitory domain Apc1-loop$^{300}$ is dislocated from the C-box binding site, allowing Cdc20 access and APC/C activation. Thus, the Apc1-loop$^{300}$ serves as a phosphorylation-dependent molecular switch for mitotic APC/C. Intriguingly, phosphorylation of Apc1-loop$^{300}$ is

Cell Cycle Control Group, UCL Cancer Institute, University College London, London, UK
*Corresponding author. Tel: +44 020 7679 6498; E-mail: h.yamano@ucl.ac.uk

accelerated by phosphorylation of a distant subunit, Apc3 (Apc3-loop). Cdk1-dependent Apc3-loop phosphorylation recruits Cdk1-cyclinB-p9/Cks2, and then, Cdk1 phosphorylates Apc1-loop$^{300}$ in a phosphorylation relay. Thus, Apc3-loop functions as a scaffold to recruit Cdk1 on APC/C.

Another loop domain of Apc1 (Apc1-loop$^{500}$, 515–584 in *Xenopus*) was also studied in the mitotic phospho-regulation of the APC/C (Fujimitsu & Yamano, 2020). PP2A is a serine/threonine protein phosphatase that is a ternary complex consisting of a scaffold subunit A, a catalytic subunit C and regulatory subunit B (Janssens *et al*, 2008; Shi, 2009). The B subunit family is classified into four subfamilies B/B55, B′/B56, B″/PR72 and B‴/STRN, which determine specific functions of PP2A. Recent studies have revealed that B56 can recognise a short linear motif (SLiM), LxxIxE, in which Glu (E) at position 6 is crucial, but position 1 (Leu) and position 4 (Ile) can be replaced by other hydrophobic residues (Hertz *et al*, 2016; Wang *et al*, 2016; Wu *et al*, 2017). Phosphorylation of residue(s) at position 2 or 3 was also reported to increase the affinity of PP2A-B56 for the SLiM. We have recently reported that Apc1-loop$^{500}$ contains a B56-binding motif and binds to PP2A-B56 in mitosis, which in turn dephosphorylates Cdc20 and controls its loading for APC/C activation (Fujimitsu & Yamano, 2020). Notably, the key Cdk1 sites around the C-box of Cdc20 are threonine (Labit *et al*, 2012; Hein *et al*, 2017; Lara-Gonzalez *et al*, 2019), whereas Cdk1-phosphorylation sites in Apc1 loop$^{300}$ are exclusively serine. As PP2A complexes have an inherent preference for phosphothreonine over phosphoserine, it seems that Cdc20 can be more efficiently dephosphorylated than the APC/C, allowing APC/C$^{Cdc20}$ complex formation during the correct window. Thus, the activity of APC/C is regulated through the association of Cdk1 kinase and PP2A phosphatase with flexible loop domains on the APC/C. However, the relationship of these counteractive regulators recruited on the APC/C remains unknown.

The evolutionarily conserved polo-like kinase 1 (Plk1) is an essential serine/threonine kinase for cell cycle events and phosphorylates numerous proteins including APC/C in mitosis (van Vugt & Medema, 2005; Archambault & Glover, 2009; Schmucker & Sumara, 2014; Zitouni *et al*, 2014; Cuijpers & Vertegaal, 2018). Plk1 is structurally divided into an N-terminal kinase domain and a C-terminal polo-box domain (PBD) that is composed of two polo-boxes (Cheng *et al*, 2003; Elia *et al*, 2003b). The groove created by these two polo-boxes can interact with phosphopeptide containing S-[pS/pT]-P/X (pS/pT denotes phosphoserine or phosphothreonine) (Elia *et al*, 2003a), contributing to substrate specificity and subcellular localisation of Plk1 (Lee *et al*, 1998; Jang *et al*, 2002; Hanisch *et al*, 2006). Plk1-PBD also has a hydrophobic pocket that is located adjacent to the phosphopeptide binding groove (Jia *et al*, 2015; Zhu *et al*, 2016). The hydrophobic pocket is thought to be important for formation of a PBD-dimer and for substrate recognition by assisting phospho-dependent interaction in the groove (Jia *et al*, 2015; Zhu *et al*, 2016; Sharma *et al*, 2019). Plk1 can facilitate the activation of APC/C in the presence of CDK activity *in vitro* (Golan *et al*, 2002; Kraft *et al*, 2003). Many Plk1-phosphorylation sites are identified on APC/C in the presence or absence of CDK (Kraft *et al*, 2003; Zhang *et al*, 2016a), yet a detailed mechanism for Plk1 regulation of APC/C activation remains unclear.

Here, we show that *Xenopus* Plx1 (Plk1 homolog) binds to Apc1-loop$^{500}$ in mitosis and promotes APC/C-Cdc20 complex formation.

We also demonstrate how Plx1 and PP2A-B56 loading to Apc1-loop$^{500}$ are coordinated through its site-specific phosphorylation, which is important for rapid entry and exit from mitosis. This study reveals the mechanism of dynamic control of the APC/C by polo-like kinase and PP2A-B56 phosphatase and underscores the importance of flexible loop domains.

## Results

### Polo-like kinase binds to Apc1-loop$^{500}$ in a phosphorylation-dependent manner

We found that a flexible loop domain of Apc1 [Apc1-loop$^{500}$: 515–584 in *Xenopus*] contains two tandem copies of the polo-binding motifs "S-[pS/pT]-[P/X]" (PBMs), in which X indicates any amino acid. These are highly conserved among species and are located upstream of a PP2A-B56-binding motif (Fig 1A). To investigate a role for these PBMs on Apc1-loop$^{500}$, we first tested whether Apc1-loop$^{500}$ binds to Plx1 (polo-like kinase homolog in *Xenopus laevis*) in *Xenopus* egg extracts. When wild-type (WT) Apc1-loop$^{500}$ fragment fused to maltose-binding protein (MBP) was incubated with anaphase extracts supplemented with non-degradable cyclin B, to ensure continuation of the anaphase state even after activation of the APC/C, endogenous Plx1 binding to Apc1-loop$^{500}$ was observed, over and above binding to MBP itself (Fig 1B). To determine the sites responsible for Plx1 binding, we introduced several CDK site mutations in the Apc1-loop$^{500}$ as shown in Fig 1A and tested their Plx1-binding activity in anaphase extracts (Fig 1B and C). The construct with T532A mutation proved to retain a similar Plx1-binding activity to WT, but T539A mutation significantly reduced the Plx1 binding, and simultaneous mutations (2A) almost abolished Plx1 binding (Fig 1C). In addition, we found that S558A mutation within the PP2A-B56-binding motif (Fujimitsu & Yamano, 2020), which is more than 20 residues downstream of PBMs, dramatically reduced Plx1 binding. S558 may be involved in Plx1 binding in an unique manner (See Discussion). Unlike mitotic extracts, the interaction between Plx1 and Apc1-loop$^{500}$ was not observed in interphase extracts, suggesting the interaction depends on phosphorylation (Appendix Fig S1A and B).

Polo-like kinase comprises a N-terminal kinase domain and a C-terminal PBD that is responsible for recognition of phosphoserine/threonine in the PBM. To investigate whether the PBD is important for its binding to Apc1-loop$^{500}$, we made Plx1-PBD and tested the ability to bind to MBP-Apc1-loop$^{500}$ constructs in anaphase extracts (Fig 1D). Plx1-PBD was co-purified with Apc1-loop$^{500}$ (WT), whereas the mutations within PBMs (T532A/T539A: 2A) abolished PBD binding (Fig 1D and E), consistent with the binding of endogenous Plx1 to Apc1-loop$^{500}$. To further confirm that Plx1 interaction with Apc1-loop$^{500}$ is through the PBD, we introduced the Pincer mutation H532A/K534M into Plx1-PBD, which abrogates the recognition of phosphopeptides (Elia *et al*, 2003b). Unlike PBD-WT, Apc1-loop$^{500}$-bound Plx1-PBD Pincer (Pin) remained at nearly undetectable levels even if a higher concentration of PBD-Pin was used (Figs 1F and EV1). In addition, our stoichiometry analysis revealed that more than one molecule of Plx1 bound per Apc1-loop$^{500}$ molecule (Fig EV1). Finally, we reconstituted the interaction between Plx1-PBD and Apc1-loop$^{500}$ using purified Cdk2-cyclin A, Plx1-PBD

and Apc1-loop[500] (Fig 1G and Appendix Fig S2). The interaction between Plx1-PBD and Apc1-loop[500] is Cdk-dependent and regulated by phosphorylation of T539 and S558. Altogether, these results indicate that Plx1 binds to Apc1-loop[500] via its PBD in a phosphorylation-dependent manner.

## The binding of Plx1 to Apc1-loop[500] promotes APC/C-Cdc20 complex formation

Plx1 has been reported to stimulate the activity of the APC/C ubiquitin ligase, yet the underlying mechanism is still elusive. To

investigate whether the binding of Plx1 to Apc1-loop[500] stimulates APC/C activity, we made a series of mutant APC/Cs with mutations in Apc1-loop[500] using insect cells and the MultiBac system (Berger *et al*, 2004; Zhang *et al*, 2013; Fujimitsu *et al*, 2016) and assessed the activities of the purified mutant APC/Cs. We examined the abilities of the mutant APC/Cs to bind Cdc20 in *Xenopus* egg extracts depleted of endogenous APC/C. In interphase extracts, all the tested APC/Cs showed a very low almost background level of Cdc20 binding (Fig 2A, lanes 13–17, and Fig 2B). In contrast, in anaphase extracts, WT APC/C bound Cdc20 to form APC/C-Cdc20 complex (Fig 2A, lane 18, and Fig 2B). However, mutations within Apc1-

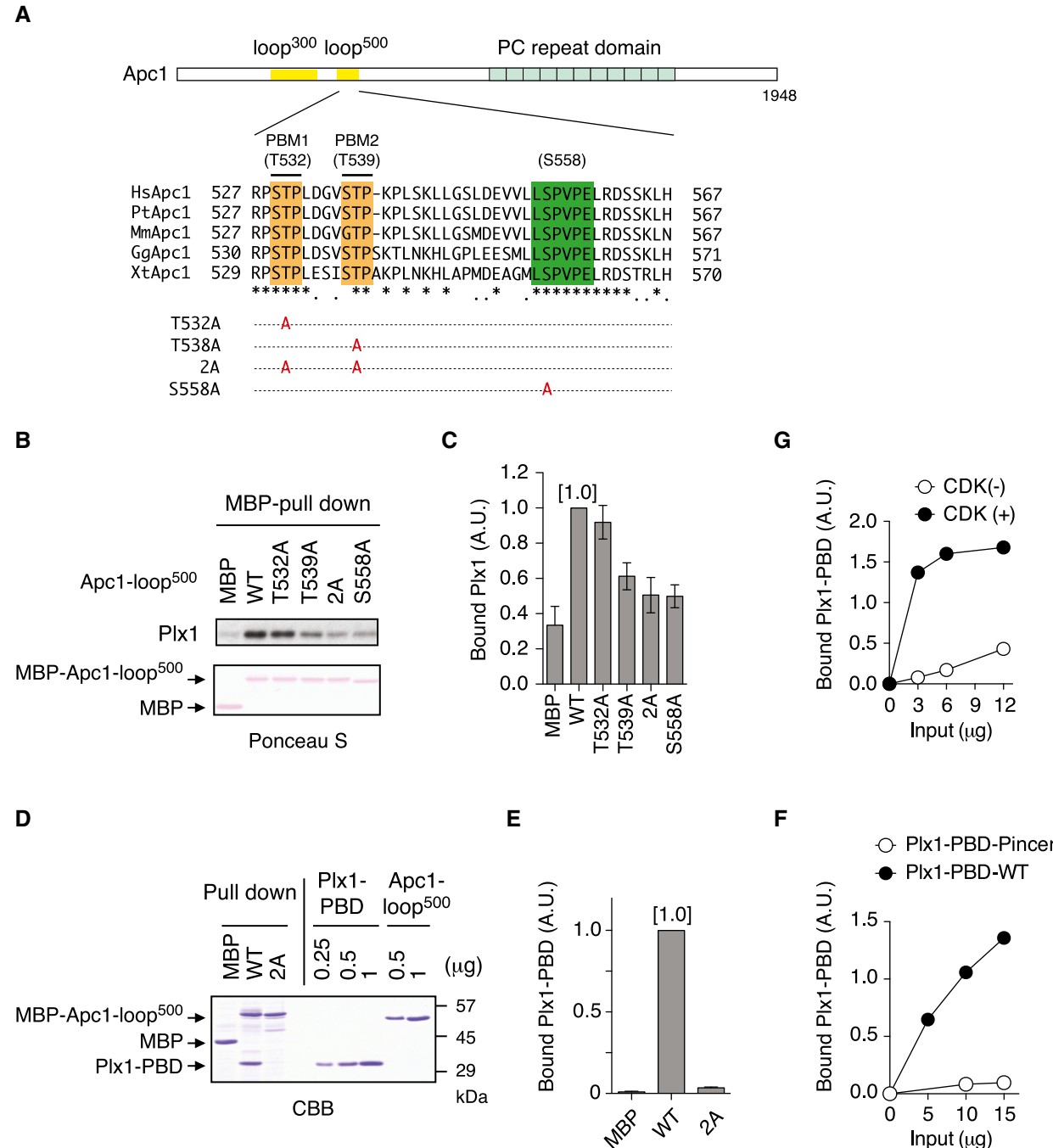

**Figure 1.**

◀

**Figure 1.  Plx1 binds to Apc1-loop$^{500}$ via its polo-box domain.**

A   Schematic diagram of Apc1 is shown. Two loop domains (loop$^{300}$, 294–399; loop$^{500}$, 515–584) and PC repeat domain are shown in yellow and in cyan, respectively. Multiple alignment of sequences of Apc1-loop$^{500}$ containing PBMs (Orange) and B56-binding site (green) is shown. Introduced alanine substitutions are indicated as A in red. Hs, *Homo sapiens* human; Pt, *Pan troglodytes* chimpanzee; Mm, *Mus musculus* mouse; Gg, *Gallus gallus* chicken; and Xt, *Xenopus tropicalis* frog. Conserved or similar amino acids are shown with an asterisk (*) or dot (.), respectively.

B   Binding assay using MBP-fused Apc1-loop$^{500}$ fragments. MBP-fused Apc1-loop$^{500}$ WT or its derivatives (T532A, T539A, 2A (T532A/T539A) and S558A) was incubated with anaphase extracts supplemented with non-degradable cyclin B at 23°C for 1 h. The bound proteins were recovered by amylose beads, separated by SDS–PAGE and detected by immunoblotting with Plx1 antibody and Ponceau S staining.

C   Quantification of (B). The bar graph is quantification of bound Plx1. The intensities of Apc1-loop$^{500}$ WT were arbitrarily set to 1.0. Error bars, SEM from three independent experiments.

D   Specific binding of Plx1-PBD to Apc1-loop$^{500}$. MBP-fused Apc1-loop$^{500}$ WT or its derivatives 2A (T532A/T539A) was incubated in anaphase extracts supplemented with non-degradable cyclin B at 23°C for 1 h and further incubated in the presence of 10 μg WT Plx1-PBD for 15 min. The bound proteins were recovered by amylose beads, separated by SDS–PAGE and detected by Coomassie brilliant blue staining (CBB). Purified proteins (Plx1-PBD and Apc1-loop$^{500}$) were run as a standard.

E   Quantification of (D). The bar graph is quantification of bound Plx1-PBD. The intensities of WT were arbitrarily set to 1.0. Error bars, SEM from three independent experiments.

F   Plx1-PBD, but not its Pincer mutant, binds to Apc1-loop$^{500}$. MBP-fused Apc1-loop$^{500}$ WT was incubated in anaphase extracts supplemented with non-degradable cyclin B at 23°C for 1 h and further incubated in the presence of indicated amounts of WT Plx1-PBD or Plx1-PBD Pincer mutant (Pin) for 15 min. The bound proteins were recovered by amylose beads and analysed as described in (D). The bound Plx1-PBD is shown as stoichiometries (the bound Plx1-PBD per Apc1-loop$^{500}$) determined from Fig EV1.

G   CDK-dependent binding of Plx1-PBD to the Apc1-loop$^{500}$. MBP-fused Apc1-loop$^{500}$ was incubated in the presence or absence of purified Cdk2/cyclin A at 30°C for 2 h. MBP-fused Apc1-loop$^{500}$ (CDK$^{+/-}$) was isolated by amylose beads and further incubated with indicated amounts of WT Plx1-PBD on ice for 10 min. The bound proteins were recovered by amylose beads and analysed as described in (D).

loop$^{500}$ impaired Cdc20 recruitment (Fig 2B), which correlates with Plx1-binding ability (Fig 1C). Apc1-T539A mutation (hereafter as 1-T539A) had a stronger impact on APC/C-Cdc20 complex formation than 1-T532A mutation. As expected, combined mutations (1–2A) significantly inhibited APC/C-Cdc20 complex formation (Fig 2A, lane 21, and Fig 2B). A similar result was obtained for the mutant APC/C with 1-S558A mutation (Fig 2A, lane 22, and Fig 2B). The mutant APC/Cs deficient in Plx1 binding (1–2A and 1-S558A) showed reduced activities in a reconstituted ubiquitin assay (Fig 2C and Appendix Fig S3). Consistently, Cdc20-dependent cyclin destruction with these mutant APC/Cs was slower than that with WT APC/C in anaphase extracts (Fig EV2). As a control, we examined cyclin destruction in interphase extracts supplemented with Cdh1 and either WT APC/C or APC/C (1–2A and 1-S558A). We found that there was no difference in the rate of cyclin destruction in interphase, suggesting that all reconstituted APC/Cs were equally functional. These results suggest that Plx1 binding on Apc1-loop$^{500}$ promotes APC/C-Cdc20 complex formation in anaphase and thereby activation of the APC/C.

We wanted to investigate how Plx1 binding to Apc1-loop$^{500}$ stimulates APC/C-Cdc20 complex formation. During the analysis of mutant APC/Cs in anaphase, we repeatedly found that the APC/C with mutations (1-T539, 1–2A and 1-S558A) displayed a decreased band shift of Apc3 in anaphase (Fig 2A, lanes 8–12), implying that the Plx1 binding on Apc1-loop$^{500}$ promotes phosphorylation of Apc3. Thus, we performed a time-course experiment during mitotic progression from interphase. Phosphorylation of Apc3 was delayed by approximately 10 min in the mutant APC/Cs (Appendix Fig S4). An extended flexible loop domain in Apc3 (Apc3-loop; 182–451 in *X. laevis* Apc3) was shown to promote activation of APC/C by stimulating phosphorylation of another loop domain of Apc1 (Apc1-loop300; 298–399 in *Xenopus* Apc1) (Fujimitsu *et al*, 2016). Therefore, we asked whether Plx1 can directly phosphorylate Apc3-loop using purified proteins. When the Apc3-loop fragment was incubated with purified Plx1, the Apc3-loop fragment showed band shifts (Fig 2D), suggesting that Plx1 can directly phosphorylate

Apc3-loop. This was confirmed by *in vitro* phosphorylation assay using [$^{32}$P]-ATP (Fig 2E and F). Apc3-loop seems the best substrate for Plx1 among four examined. It is also noteworthy that a mutant Apc3-loop with mutations in nine CDK sites (Apc3-loop-9A: T206A, S242A, S277A, T280A, T290A, S292A, T343A, T360A and S365A) was phosphorylated by Plx1 at a high level (Fig 2F), suggesting that Plx1 mainly phosphorylates different sites from these CDK sites in Apc3-loop.

### A crosstalk between Plx1 and PP2A-B56 on Apc1-loop$^{500}$

Apc1-loop$^{500}$ was shown to bind PP2A-B56, which promotes formation of APC/C-Cdc20 complex through dephosphorylation of Cdc20 (Fujimitsu & Yamano, 2020). Thus, Apc1-loop$^{500}$ is the hub for both a kinase and a phosphatase, Plx1 and PP2A-B56, respectively. We wanted to investigate the interplay between Plx1 and PP2A-B56 (Fig 3A). To investigate whether the association of PP2A-B56 with Apc1-loop$^{500}$ can regulate the phosphorylation of PBMs, we made MBP-fused Apc1-loop$^{500}$ fragments carrying mutations in the B56-binding motif (L557A/V560A or E562A; Fig EV3A). As expected, these mutations inhibited the binding of B56 to Apc1-loop$^{500}$ in anaphase extracts at a similar level (Fig EV3B and C) (Fujimitsu & Yamano, 2020). To assess the impact of PP2A-B56 association on Plx1 binding, we then examined phosphorylation states of CDK sites (T532 and T539) within the PBMs using phospho-site-specific antibodies (Fig 3B). Phosphorylation levels of T532 and T539 were stimulated by both of these mutations (L557A/V560A and E562A). These results suggest that the PP2A-B56 associated with Apc1-loop$^{500}$ promotes dephosphorylation of T532 and T539 within the PBMs. The specificity of the phospho-specific antibodies was verified in Fig EV4, which also highlights how each CDK site mutation influences the phosphorylation status of other CDK sites within Apc1-loop$^{500}$ in anaphase extracts (Fig EV4A). As Plx1 binding to Apc1-loop$^{500}$ is dependent on phosphorylation, we also examined the Plx1-binding activity of these Apc1-loop mutant fragments. Surprisingly, these two mutations had different effects on Plx1

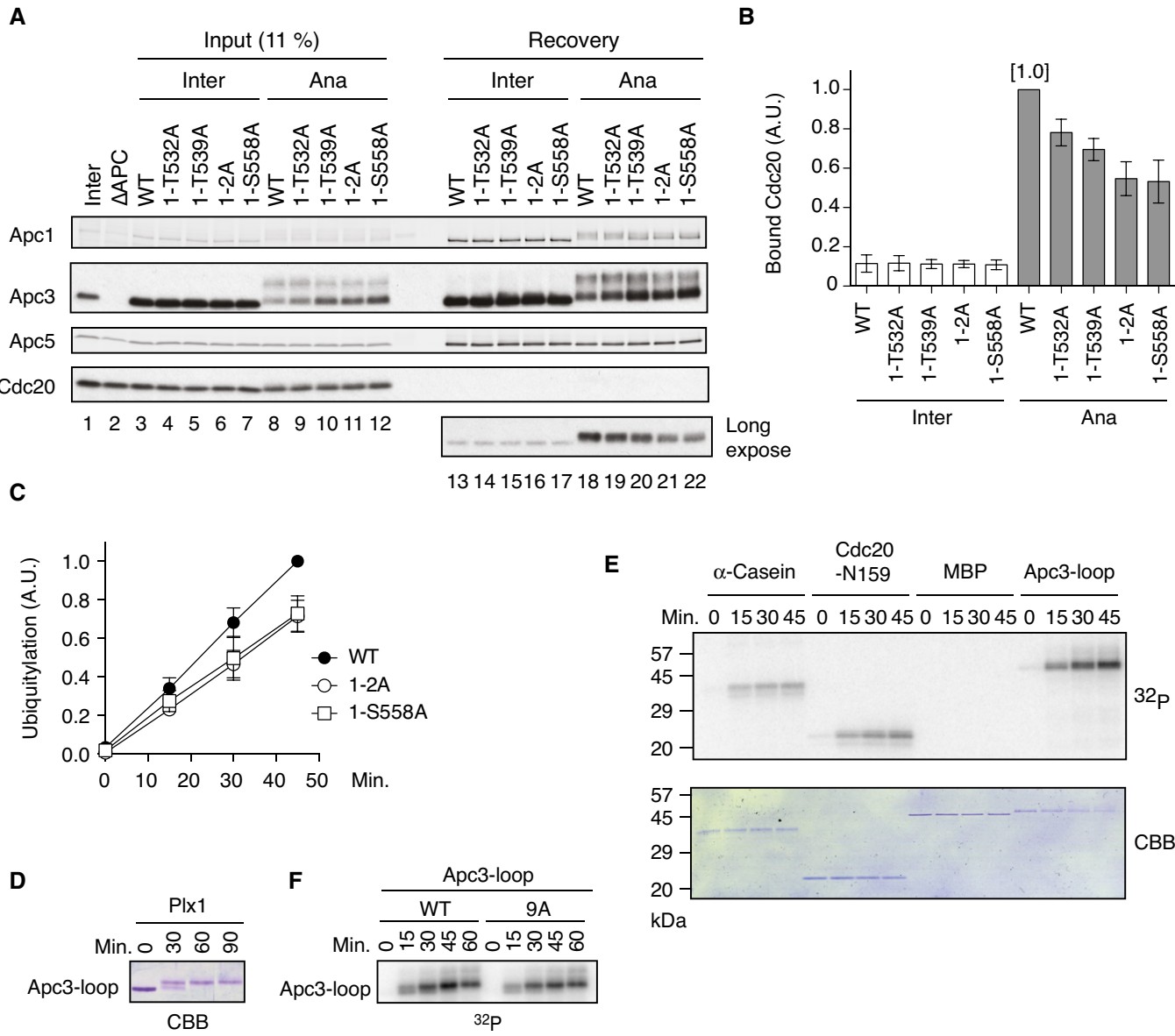

**Figure 2. Plx1-bound Apc1 promotes APC/C-Cdc20 complex formation.**

A   Cdc20-binding assay in *Xenopus* egg extracts. The purified recombinant WT or Apc-loop[500] mutant APC/Cs with mutations (1-T532A, 1-T539A, 1-T532A/T539A (1–2A) and 1-S558A) was incubated with APC/C-depleted (ΔAPC) interphase extract (Inter) or ΔAPC anaphase extracts supplemented with non-degradable cyclin B (Ana) at 23°C for 1 h. The APC/C was recovered with Apc3 monoclonal antibody (AF3.1) beads, and the bound proteins were analysed by SDS–PAGE and immunoblotting with indicated antibodies.

B   Quantification of (A). The bar graph is quantification of bound Cdc20. The intensities of WT control in anaphase were arbitrarily set to 1.0. Error bars, SEM from three independent experiments.

C   Plx1-binding-deficient APC/Cs are less active in ubiquitylation assay than WT APC/C. The purified recombinant WT APC/C or mutant APC/C (1-2A or 1-S558A) was incubated with ΔAPC anaphase extract and the recovered APC/C-Cdc20 complex was subjected to ubiquitylation assay (Appendix Fig S3). Ubiquitylation was quantified, and the activity of WT at 45 min was arbitrarily set to 1.0. Error bars, SEM from three independent experiments. A representative result is shown in Appendix Fig S3.

D   *In vitro* kinase assay of Apc3-loop using purified Plx1. The purified Apc3-loop (182–451 in *Xenopus*) fragment was incubated with Plx1 in the presence of ATP at 23°C for indicated times, separated by SDS–PAGE and detected by Coomassie brilliant blue (CBB).

E   *In vitro* phosphorylation by Plx1. Apc3-loop fragment, α-Casein, N-terminus of Cdc20 (N159) or MBP protein was incubated with purified Plx1 in the presence of [γ32P]-ATP at 23°C for indicated times, separated by SDS–PAGE and detected by autoradiography and Coomassie brilliant blue (CBB) staining.

F   Phosphorylation of Apc3-loop by Plx1 in the presence of [γ32P]-ATP. WT Apc3-loop fragment or its CDK site mutant (9A) was incubated with purified Plx1 in the presence of [γ32P]-ATP at 23°C for indicated times, separated by SDS–PAGE and detected by autoradiography.

binding (Fig 3C and D). E562A mutation increased the Plx1 binding to Apc1-loop$^{500}$, which is likely due to an increased phosphorylation of the PBMs (at T532 and T539). In contrast, L557A/V560A mutation inhibited the binding of Plx1. Either L557 or V560 or both residues may be required for Plx1 binding as these two residues are close to the S558 site that is required for Plx1 binding (Fig 1B and C) as well as B56 binding (Fujimitsu & Yamano, 2020).

To further investigate the relationship between Plx1 and B56 subunit, we investigated the impacts of the mutations on Apc1-loop$^{500}$ (Fig 1A) on B56-binding activity in anaphase extracts (Fig 3E and F). Intriguingly, T532A mutation, which barely had an impact on Plx1 binding (Fig 1C and Appendix Fig S2B), abolished B56 binding (Fig 3F), whereas Apc1-loop$^{500}$ with the T539A mutation bound B56γ equally well as WT. These results suggest that B56 binding and Plx1 binding show different phosphorylation-site dependency on Apc1-loop$^{500}$, T532 for PP2A-B56 loading and T539 for Plx1, respectively (Fig 3G). Acidic residues or phosphorylation in the region downstream of the B56-binding motif can enhance the affinity to the B56 regulatory subunit (Hertz *et al*, 2016; Wang *et al*, 2016). We therefore investigated whether phosphorylation of S566/T567, which is located 4 residues downstream of the B56-binding motif and is a possible Plx1-binding site, is involved in Plx1 or B56 binding (Appendix Fig S5). Alanine mutations in S566/T567 (S566A/T567A) had no impact on Plx1 or B56 loading, suggesting that phosphorylation of S566/T567 is not required for either B56 or Plx1 binding to Apc1-loop$^{500}$.

## Plx1 and PP2A-B56 compete for binding to Apc1-loop$^{500}$

The mutations (S558A and L557A/V560A) in PP2A-B56-binding site strongly inhibited not only PP2A-B56 binding but also Plx1 binding. We speculated that Plx1 and PP2A-B56 might share the same binding region presumably around S558. To verify this idea, we investigated whether Plx1 was able to compete with PP2A-B56 for its binding to Apc1-loop$^{500}$. When MBP-fused WT Apc1-loop$^{500}$ and $^{35}$S-labelled B56 subunit were incubated in anaphase extracts, B56 binding to Apc1-loop$^{500}$ was clearly observed by pull-down assays (Fig 4A, upper panel, lane 10). However, when WT Plx1-PBD was added to the reaction as a competitor, B56 binding was reduced to a

background level similar to that of B56 binding to the Δ11 mutant Apc1-loop$^{500}$ (Δ11, deletion of 11 residues including B56-binding motif; Fig 4A, upper panel, lanes 11–14). At the same time, the binding of Plx1-PBD to Apc1-loop$^{500}$ (WT) was observed (Fig 4A lower panel, lanes 12–14), suggesting that PP2A-B56 on Apc1-loop$^{500}$ was replaced by Plx1-PBD. In contrast, a mutant Plx1-PBD (Pincer) harbouring a mutation by which Plx1 is unable to bind PBMs failed to inhibit B56 binding to Apc1-loop$^{500}$ (Fig 4A, upper panel, lanes 15 and 16). These results suggest that Plx1 and PP2A-B56 share the same binding region in Apc1-loop$^{500}$.

Given that Plx1 binds to phosphorylated Apc1-loop$^{500}$, we hypothesised that PP2A-B56 might promote dissociation of Plx1 from Apc1-loop$^{500}$ through dephosphorylation of Apc1-loop$^{500}$. We first sought to confirm that purified PP2A-B56γ dephosphorylates phosphorylated T532, T539 and S558 (pT532, pT539, pS558) in Apc1-loop$^{500}$ (Fig 4B). MBP-fused Apc1-loop$^{500}$ fragment was phosphorylated in anaphase extracts and isolated using amylose beads, and then incubated with purified PP2A-B56γ. When WT Apc1-loop$^{500}$ was used as a substrate, T532 was dephosphorylated by purified PP2A-B56γ (Fig 4B, lanes 2 and 3). In contrast, when the same loop fragment with mutations on the B56-binding site (L557A/V560A and E562A) was used, dephosphorylation of T532 was reduced. Surprisingly, T539 of WT Apc1-loop$^{500}$ was dephosphorylated in the absence of purified PP2A-B56γ (Fig 4B, lanes 2 and 3), whereas dephosphorylation of T539 of B56-binding site mutants (L557A/V560A and E562A) was dependent on the addition of purified PP2A-B56γ (Fig 4B, lanes 5, 6, 8 and 9). This is most likely caused by co-purified PP2A-B56 with Apc1-loop$^{500}$ from anaphase extracts, highlighting that PP2A-B56 associated with Apc1-loop$^{500}$ preferentially dephosphorylates T539 over T532. In agreement with a recent report that an SP site is a particularly poor substrate for PP2A-B56 (Kruse *et al*, 2020), S558 was a poor substrate for PP2A-B56γ. These results suggest that PP2A-B56γ can dephosphorylate T532, T539 and S558 in Apc1-loop$^{500}$ with a distinct site preference. Furthermore, we examined how Plx1 binding to Apc1-loop$^{500}$ influenced PP2A-B56-dependent dephosphorylation. Intriguingly, in the presence of Plx1-PBD, Apc1-loop$^{500}$ fragment became a poorer substrate for PP2A-B56 than in its absence (Fig 4C), highlighting that Plx1 binding had an inhibitory effect on dephosphorylation of

**Figure 3. Interplay between Plx1 and PP2A-B56 on Apc1-loop$^{500}$.**

A   Apc1-loop$^{500}$ is the hub for both Plx1 and PP2A-B56 phosphatase. CDK sites (T or S) and B56-binding site (green) on Apc1-loop$^{500}$ (yellow bar) are shown. Bindings of Plx1 and B56 to Apc1-loop$^{500}$ depend on phosphorylation of CDK sites. The PP2A-B56 associated with Apc1-loop$^{500}$ may dephosphorylate CDK sites.

B   Phosphorylation status of T532 and T539 in Apc1-loop$^{500}$ varies depending on mutations in the B56-binding site. MBP-fused Apc1-loop$^{500}$ WT or its derivatives (L557A/V560A and E562A) was incubated with anaphase extracts supplemented with non-degradable cyclin B at 23°C for 1 h. The proteins were recovered by amylose beads, separated by SDS–PAGE and detected by immunoblotting with phospho-specific (pAb) or MBP antibody.

C   Plx1-binding assay using MBP-fused Apc1-loop$^{500}$ fragments with mutations in B56-binding site. The Plx1 binding to Apc1-loop$^{500}$ WT or its derivatives (L557A/V560A or E562A) was analysed as described in Fig 1B.

D   Quantification of (C). The bar graph is quantification of bound Plx1. The intensities of WT were arbitrarily set to 1.0. Error bars, SEM from three independent experiments.

E   Binding assay using MBP-fused Apc1-loop$^{500}$ fragments and B56γ MBP-fused Apc1-loop$^{500}$ WT or its derivatives (T532A, T539A, T532A/T539A (2A) and S558A) was incubated with the $^{35}$S-labelled Flag-B56γ in anaphase extracts supplemented with non-degradable cyclin B at 23°C for 1 h. The bound proteins were recovered by amylose beads, separated by SDS–PAGE and detected by autoradiography or Coomassie brilliant blue (CBB) staining. The recovery of WT (×1/2 and ×1) was run as a standard.

F   Quantification of (E). The bar graph is quantification of bound B56γ. The intensities of WT were arbitrarily set to 1.0. Error bars, SEM from three independent experiments.

G   Summary of Plx1 and B56 binding to Apc1-loop$^{500}$ derivatives. The abilities of Apc1-loop$^{500}$ fragments carrying the indicated mutations are shown. The B56 bindings to Δ11 (deletion of 11 residues including B56-binding site) and 3A (T532A/T539A/S558A) were tested previously (Fujimitsu & Yamano, 2020).

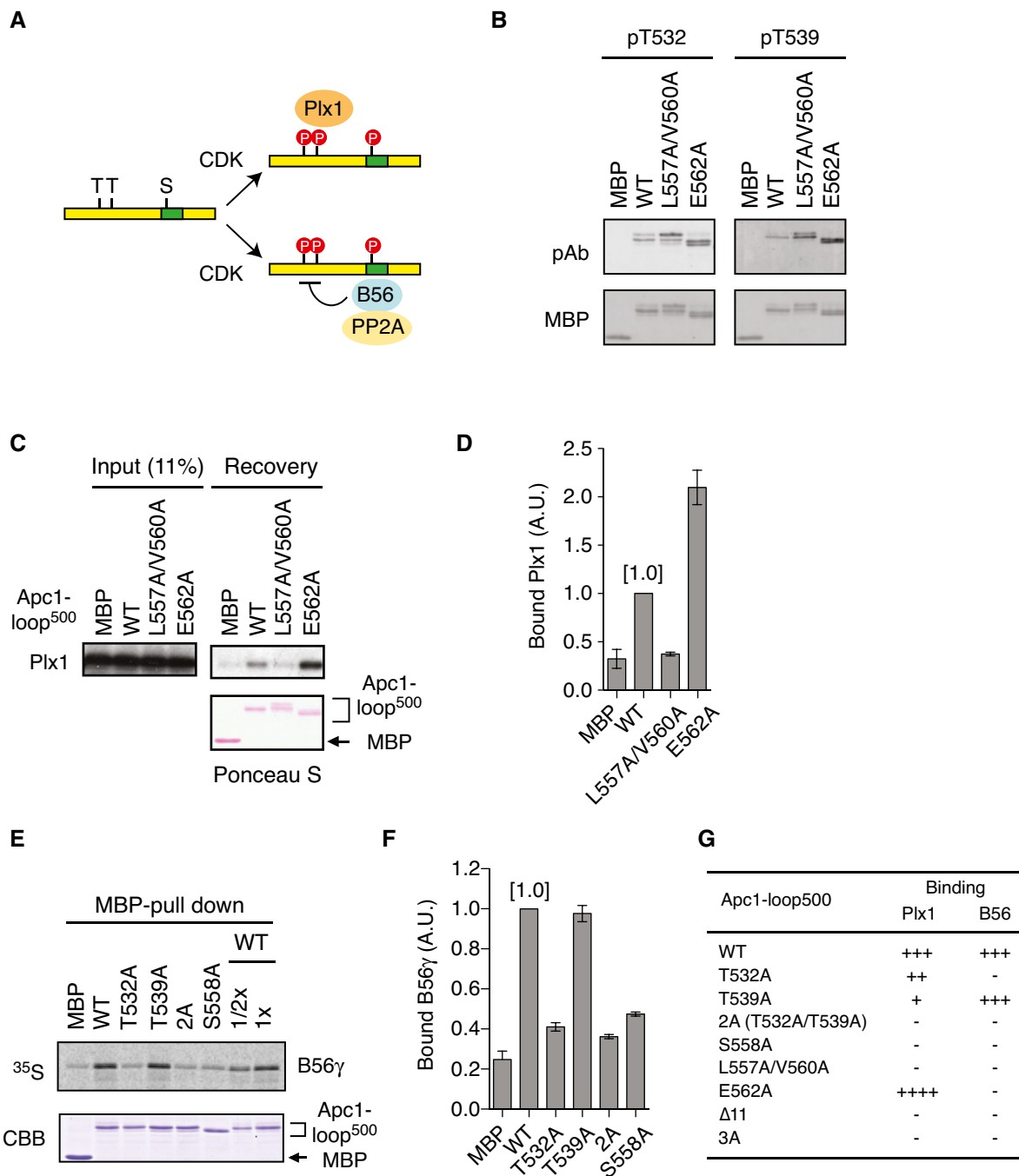

Figure 3.

T532, T539 and S558. These results support the idea that Plx1 and PP2A-B56 compete for binding to Apc1-loop$^{500}$ through phospho-regulation as well as sharing binding sites.

**T539 is phosphorylated at earlier timing than T532 in anaphase**

The phosphorylation status of each site of the protein is controlled by the cooperative action of both kinases and phosphatases responsible for the site. As B56 binding is apparently dependent upon phosphorylation of T532 (pT532) whereas Plx1-bindng is dependent upon pT539, we wanted to investigate how these CDK sites on Apc1-loop$^{500}$ were phosphorylated in anaphase. Firstly, phosphory-lation status of T532, T539 and S558 on MBP-Apc1-loop$^{500}$ was monitored in anaphase extracts by phospho-site-specific antibodies. At time 0, when MBP-Apc1-loop$^{500}$ was added into interphase extracts, no band was detected by any of the phospho-site-specific antibodies used, indicating that the sites are recognised only when they are phosphorylated (Fig 5A). Intriguingly, S558 and T539 sites were both more rapidly phosphorylated than T532 (Fig 5A and B), enabling the loading of Plx1 to Apc1-loop$^{500}$ (Fig 1B and C). This

was true even in the context of the APC/C. T532 of Apc1-loop$^{500}$ is phosphorylated more slowly than T539 and S558 within the APC/C (Figs 5C and EV5). Eventually, when T532 is sufficiently phosphorylated, PP2A-B56 loading to Apc1-loop$^{500}$ takes place. We further examined whether PP2A-B56γ promoted dissociation of Plx1-PBD from Apc1-loop$^{500}$ fragment (Fig 5D). As shown in Fig 1E, Plx1-PBD efficiently binds to Apc1-loop$^{500}$ in anaphase, and thus, this complex was isolated and incubated with purified PP2A-B56γ. Within 60 min, nearly 50% of bound Plx1 was dissociated from WT Apc1-loop$^{500}$ in the presence of PP2A-B56γ (Fig 5D). Apc1-loop$^{500}$ harbouring E562A mutation was used as a control as this mutation specifically blocks PP2A-B56 binding but not Plx1 binding to Apc1-loop$^{500}$. Strikingly, the point mutation E562A strongly suppressed Plx1-PBD dissociation from Apc1-loop$^{500}$ (Fig 5D). This is consistent with our finding that dephosphorylation of T539, a key site for Plx1 binding, is less efficient on the Apc1-loop$^{500}$ fragment with the E562A mutation than WT (Fig 4C), presumably because Plx1 prevents PP2A-B56 loading. These results suggest that Apc1-loop$^{500}$ plays a role on dynamic regulation of the loading of Plx1 and PP2A-B56 in mitosis, Plx1 first and then later PP2A-B56.

## Overloading of Plx1 on Apc1-loop$^{500}$ accelerates the activation of the APC/C

We wanted to investigate the physiological significance of Plx1 loading on to Apc1-loop$^{500}$ because the E562A mutation in Apc1 (1-E562A) specifically recruits Plx1, but not PP2A-B56 on Apc1-loop$^{500}$. When the mutant APC/C (1-E562A) was incubated with anaphase extracts for 45 min, the binding of Cdc20 was significantly higher than that to WT APC/C (Fig 6A, lanes 15 and 17, and Fig 6B). The increased binding of Cdc20 in the mutant APC/C (1-E562A) was moderate at 60 min (Fig 6A, lanes 18 and 20, and Fig 6B). These results suggest that 1-E562A promotes the formation of APC/C-Cdc20 complex in anaphase, which is particularly effective in an early stage. It is noteworthy that phosphorylation of Apc3 was accelerated in the mutant APC/C (1-E562A; Appendix Fig S6), which is consistent with the idea that the Plx1 binding to Apc1-loop$^{500}$ promotes phosphorylation of Apc3. As shown in a previous report (Fujimitsu & Yamano, 2020), the APC/C mutant (1-L557A/V560A) showed reduced complex formation of APC/C-Cdc20 and delayed phosphorylation of Apc3 (Fig 6A, lanes 16 and 19).

As phosphorylation of T539, a key phosphorylation site for Plx1 loading, is earlier than that of T532 (Fig 5B and C) in mitosis, we hypothesised that Plx1 might be recruited to Apc1-loop$^{500}$ in order to activate APC/C in an early stage of CDK-activation, accelerating the onset of activation of the APC/C. If this is the case, the mutant APC/C (1-E562A) might cause premature activation compared with WT APC/C. The timing of the activation of the APC/C was monitored by the timing of cyclin B destruction in anaphase extracts in which the endogenous APC/C had been replaced by WT or mutant APC/Cs. With WT APC/C, the degradation of [$^{35}$S]-labelled substrate cyclin B initiates around 30 min after the addition of non-degradable cyclin B into interphase extracts (Fig 6C, WT). However, the mutant APC/C (1-E562A) showed premature initiation of cyclin destruction (Fig 6C and Appendix Fig S7A). The mutant APC/C (1-E562A) was as active as WT APC/C in Cdh1-dependent activation in interphase (Appendix Fig S7B and C), indicating the functional integrities of these mutant APC/C complexes. Higher activity of the mutant APC/C (1-E562A) than WT APC/C in anaphase extracts was also confirmed by ubiquitylation assay (Appendix Fig S7D and E). These results indicate that Plx1 associated with Apc1-loop$^{500}$ accelerates APC/C-Cdc20 complex formation and subsequent activation of APC/C.

## Coordination of Plx1 and PP2A-B56 during mitotic exit

Dephosphorylation of the APC/C starts during mitotic exit, after the metaphase to anaphase transition triggered by CDK inactivation. To ask whether PP2A-B56 recruitment to Apc1-loop$^{500}$ is important for APC/C dephosphorylation after CDK inactivation, we monitored phosphorylation of Apc3 in the mutant APC/Cs (1-LV2A and 1-E562A) during mitotic exit by addition of p27, a CDK inhibitor. In WT APC/C, phosphorylation of Apc3 peaked at 5 min after addition of p27 (Fig 6D, lane 9), probably because there is a time lag until p27 inhibits CDK. Dephosphorylation of Apc3 appeared to start around 10 min after addition of p27 (Fig 6D, lane 12), continued and was completed by 30 min (Fig 6D, lanes 15, 18 and 21). In the mutant APC/C (1-E562A), the peak of Apc3 phosphorylation was at 10 min after addition of p27 (Fig 6D, lane 14), suggesting a slower response to inactivation of CDK. At 15 min, levels of Apc3 phosphorylation in the mutant APC/C (1-E562A) were still more prominent than those of WT APC/C even though dephosphorylation of Apc3 had already started around 10 min (Fig 6D, lanes 15 and 17). It may

---

**Figure 4. Mutual regulation of Plx1 and PP2A-B56 on Apc1-loop$^{500}$.**

A   Plx1 and PP2A-B56 competes for Apc1-loop$^{500}$. MBP-fused Apc1-loop$^{500}$ WT or its derivatives (Δ11, deletion of 11 residues including the B56-binding motif) was incubated with the $^{35}$S-labelled Flag-B56γ in anaphase extracts supplemented with non-degradable cyclin B at 23°C for 1 h and further incubated with indicated amounts of WT Plx1-PBD or Plx1-PBD Pincer mutant (Pin) for 15 min. The bound proteins were recovered by amylose beads, separated by SDS–PAGE and detected by autoradiography or Coomassie brilliant blue (CBB) staining.

B   Dephosphorylation of T532, T539 and S558 on Apc1 by PP2A-B56. MBP-fused Apc1-loop$^{500}$ WT or its derivatives (L557A/V560A and E562A) was incubated in anaphase extracts supplemented with non-degradable cyclin B at 23°C for 1 h, isolated by amylose beads and further incubated in the presence (78 nM) or absence of purified PP2A-B56γ at 23°C for 20 min. The proteins were recovered by amylose beads, separated by SDS–PAGE and detected by immunoblotting with phospho-specific or MBP antibodies. The signals of a phospho-specific antibody and MBP antibody on the same membrane were detected in the 800- and 700-nm channels, respectively. The two channels were pseudo-coloured (green; pT532, pT539 and pS558, red; MBP) and overlaid in the bottom panel (Overlay). The signal of pS558 was very faint in L557A/V560A mutant probably because these two residue are close to S558, interfering with pS558 antibody recognition.

C   Dephosphorylation of Apc1-loop$^{500}$ by PP2A-B56 in the presence of Plx1-PBD. MBP-fused Apc1-loop$^{500}$ WT or its derivatives (L557A/V560A or E562A) was incubated in anaphase extracts supplemented with non-degradable cyclin B at 23°C for 1 h and further incubated with 10 μg of WT Plx1-PBD for 15 min. The complexes were isolated by amylose beads and incubated in the presence (78 nM) or absence of purified PP2A- B56γ at 23°C for 20 min. The proteins were recovered by amylose beads and analysed as described in (B).

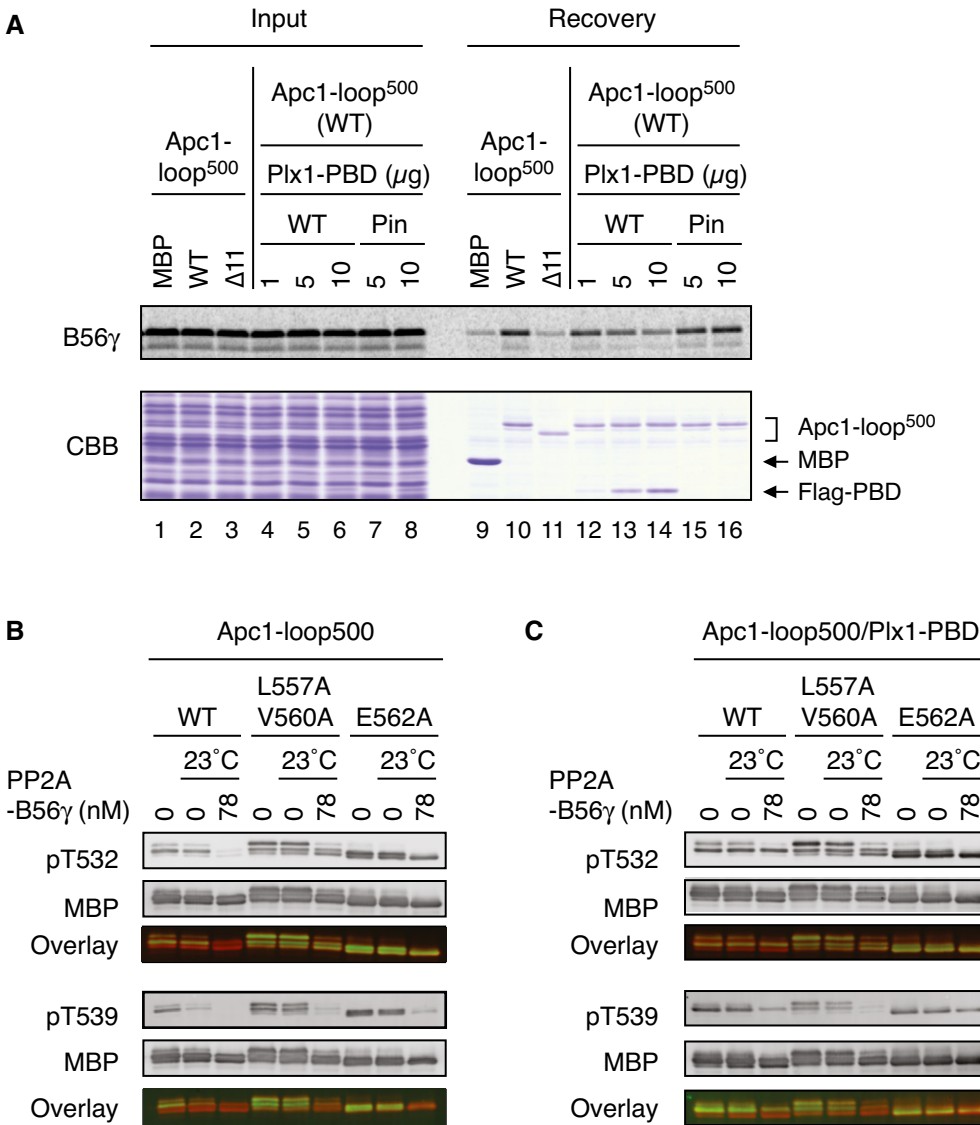

**Figure 4.**

be that overloading of Plx1 onto the APC/C reduces PP2A-B56 recruitment and decelerates dephosphorylation of the APC/C. Thus, PP2A-B56 binding to Apc1-loop[500] is important for a swift response to CDK inactivation. As expected, the mutant APC/C (1-L557A/V560A), that is unable to bind Plx1, showed less prominent Apc3 phosphorylation than WT APC/C at every time point (Fig 6D, lanes 7, 10, 13 and 16). Taken together, these results suggest that dynamic Plx1 and PP2A-B56 recruitment to Apc1-loop[500] is important not only for activation of the APC/C at the right time but also for swift APC/C dephosphorylation and inactivation as CDK is inactivated.

## A crosstalk between Apc1-loop[500] and Apc3-loop

Finally, we examined the relation between Apc1-loop[500] and Apc3-loop. We previously reported that phosphorylation of Apc1-loop[300], which is adjacent to Apc1-loop[500] in APC/C, is stimulated by the Cdk1/p9/cyclin B ternary complex loaded onto phosphorylated Apc3-loop (Fujimitsu *et al*, 2016). We therefore hypothesised that the CDK sites on Apc1-loop[500] could also be stimulated by phosphorylation of Apc3-loop by which, together with Plx1-dependent Apc3 phosphorylation, a feedback loop forms allowing dynamic APC/C

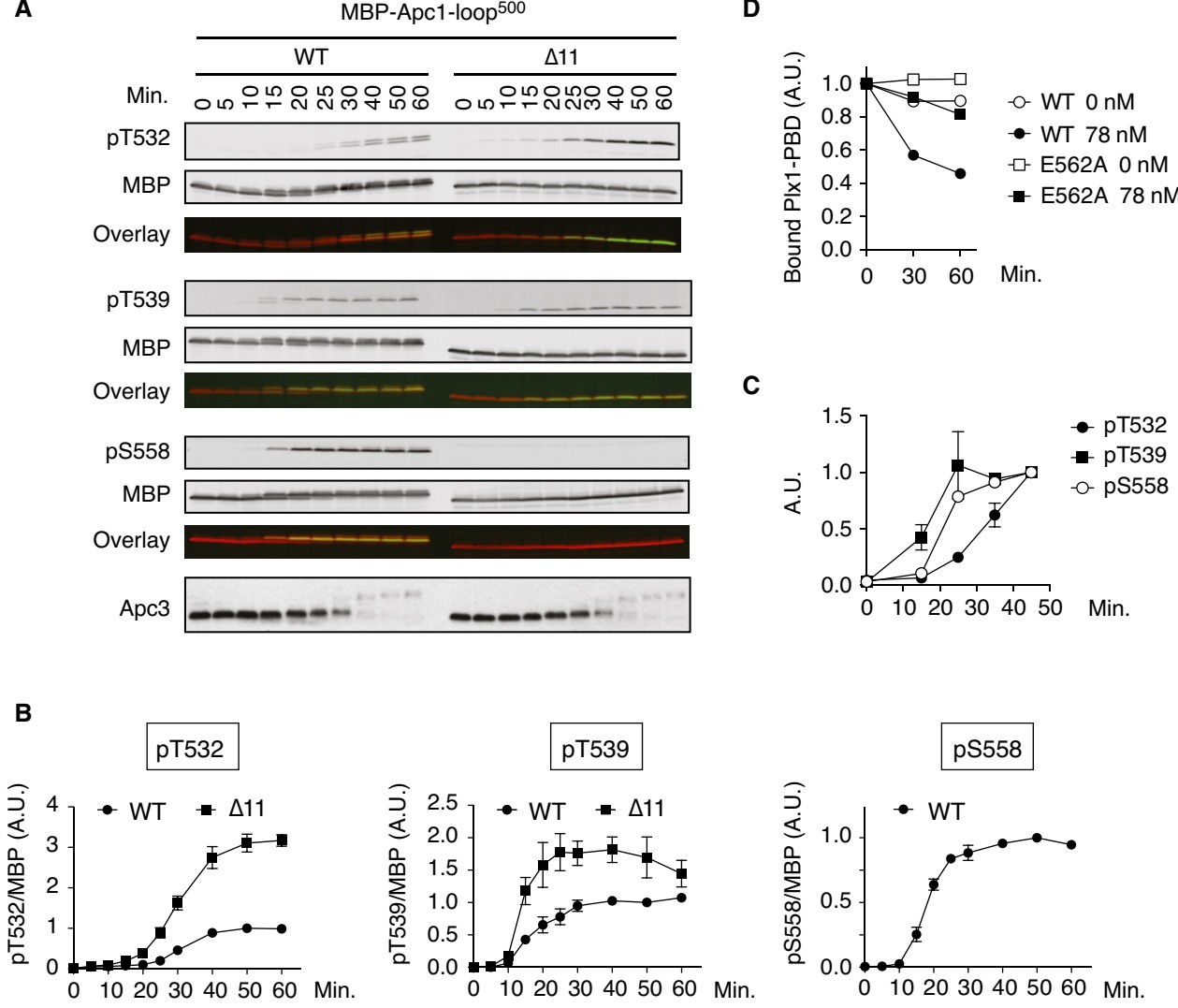

**Figure 5. Phosphorylation and function of Apc1-loop500 in Xenopus egg extracts.**

A  Phosphorylation kinetics of T532, T539 and S558 on Apc1-loop500 in anaphase extracts. MBP-fused Apc1-loop500 WT or its derivatives (Δ11, deletion of 11 residues including the B56-binding motif) was incubated with Xenopus egg extracts in the presence of non-degradable cyclin B at 23°C. The samples taken as indicated were analysed by SDS–PAGE and immunoblotting using phospho-specific antibodies (pT532, pT539 and pS558) or MBP antibody. To monitor CDK activity, Apc3 antibody was used (bottom). The signals of a phospho-specific antibody and MBP antibody on the same membrane were detected in the 800- and 700-nm channels, respectively. The two channels were pseudo-coloured (green; pT532, pT539 and pS558, red; MBP) and overlaid in the bottom panel (Overlay). The signal of pS558 was not detected in Δ11 mutant because Δ11 mutation includes S558.

B  Quantification of (A). The intensities of WT at 50 min were arbitrarily set to 1.0. Error bars, SEM from three independent experiments.

C  Phosphorylation kinetics of T532, T539 and S558 on Apc1-loop500 of the endogenous APC/C in Xenopus egg extract. Endogenous APC/C was immunoprecipitated with Apc3 MAb (AF3.1) beads at the indicated time points after mitotic induction by adding non-degradable cyclin B and analysed by SDS–PAGE and immunoblotting with phospho-specific antibodies, pT532, pT539 and pS558. Quantification of Fig EV5A. The intensities at 45 min were arbitrarily set to 1.0. Error bars, SEM from three independent experiments.

D  Dissociation of Plx1-PBD from Apc1-loop500. MBP-fused Apc1-loop500 WT or its derivatives (E562A) was incubated in anaphase extracts supplemented with non-degradable cyclin B at 23°C for 1 h and further incubated with 10 μg of WT Plx1-PBD for 15 min. The complexes were isolated by amylose beads and incubated in the presence (78 nM) or absence of purified PP2A-B56γ at 23°C for 30 or 60 min. The bound proteins were recovered by amylose beads, separated by SDS–PAGE and detected by Coomassie brilliant blue (CBB) staining. The intensities at 0 min were arbitrarily set to 1.0.

regulation during the cell cycle and in responsive to external cues. To test this possibility, we monitored phosphorylation of T532 in Apc1 (pT532), a key site for PP2A-B56 recruitment, using a specific antibody to phospho-T532, in anaphase extracts in which the endogenous APC/C had been replaced by WT or mutant APC/C.

When nine Cdk1 sites on Apc3-loop were mutated to alanine (Apc3-9A), phosphorylation of T532 on this APC/C was significantly reduced by ~ 50%, compared with that on WT APC/C (Fig 7A and B). This result indicates that phosphorylation of Apc3 stimulates phosphorylation of T532 on Apc1-loop500 and thereby initiates

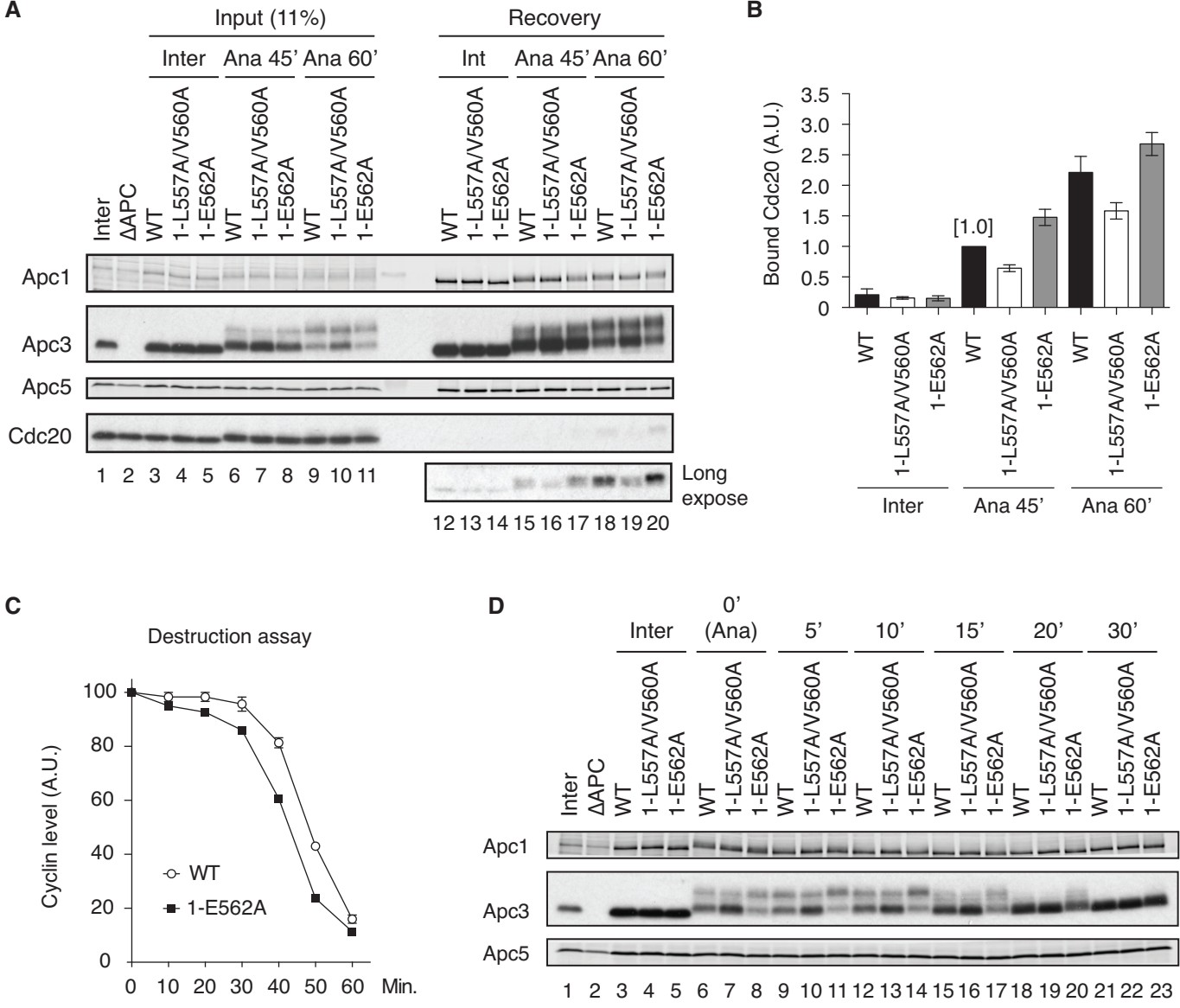

**Figure 6.  Increased Plx1 binding to Apc1-loop[500] induces premature APC/C activation and delays dephosphorylation of APC/C during mitotic exit.**

A   Cdc20-binding assay in *Xenopus* egg extracts. The Cdc20-binding activities of WT APC/C and Apc-loop[500]-mutant APC/Cs with mutations (1-L557A/V560A or 1-E562A) were analysed as described in Fig 2A.

B   Quantification of (A). The bar graph is quantification of bound Cdc20. The intensities of WT control at Ana-45 min were arbitrarily set to 1.0. Error bars, SEM from three independent experiments.

C   Premature activation of a mutant APC/C (1-E562A) in *Xenopus* egg extracts. The purified recombinant WT APC/C or Apc-loop[500]-mutant APC/C (1-E562A) was incubated with its substrates ([35]S-labelled cyclin B and a version of cyclin B lacking the N-terminal 67 residues, Δ67) in APC/C-depleted (ΔAPC) interphase extracts supplemented with non-degradable cyclin B at 23°C. Samples taken at indicated time points after addition of substrates were analysed by SDS–PAGE and autoradiography (Appendix Fig S7A). The relative cyclin levels are shown, normalised with reference to the intensities found at time 0 for each time point. Error bars, SEM from three independent experiments. A representative result is shown in Appendix Fig S7A.

D   APC/C dephosphorylation after inactivation of CDK. The purified recombinant WT or Apc-loop[500]-mutant APC/Cs with mutations (1-L557A/V560A or 1-E562A) was incubated with APC/C-depleted (ΔAPC) in anaphase extracts supplemented with non-degradable cyclin B at 23°C for 1 h (Ana). After addition of p27 (0.3 μM), the samples taken at time points shown were analysed by SDS–PAGE and immunoblotting with indicated antibodies. "Inter" denotes interphase.

PP2A-B56 recruitment. To verify the importance of T532-phosphorylation for B56 binding, we investigated the phosphorylation status of T532 and S558 of Apc1-loop[500] when PP2A-B56 was bound (Fig 7C and Appendix Fig S8). To this end, MBP-Apc1-

loop[500] was phosphorylated by Cdk2/cyclin A *in vitro* and then incubated with Flag-tagged B56γ for Apc1-loop[500] pull-down and analysed by phospho-site-specific antibodies (pT532 or pS558) as well as MBP antibodies. After phosphorylation, MBP-Apc1-loop[500]

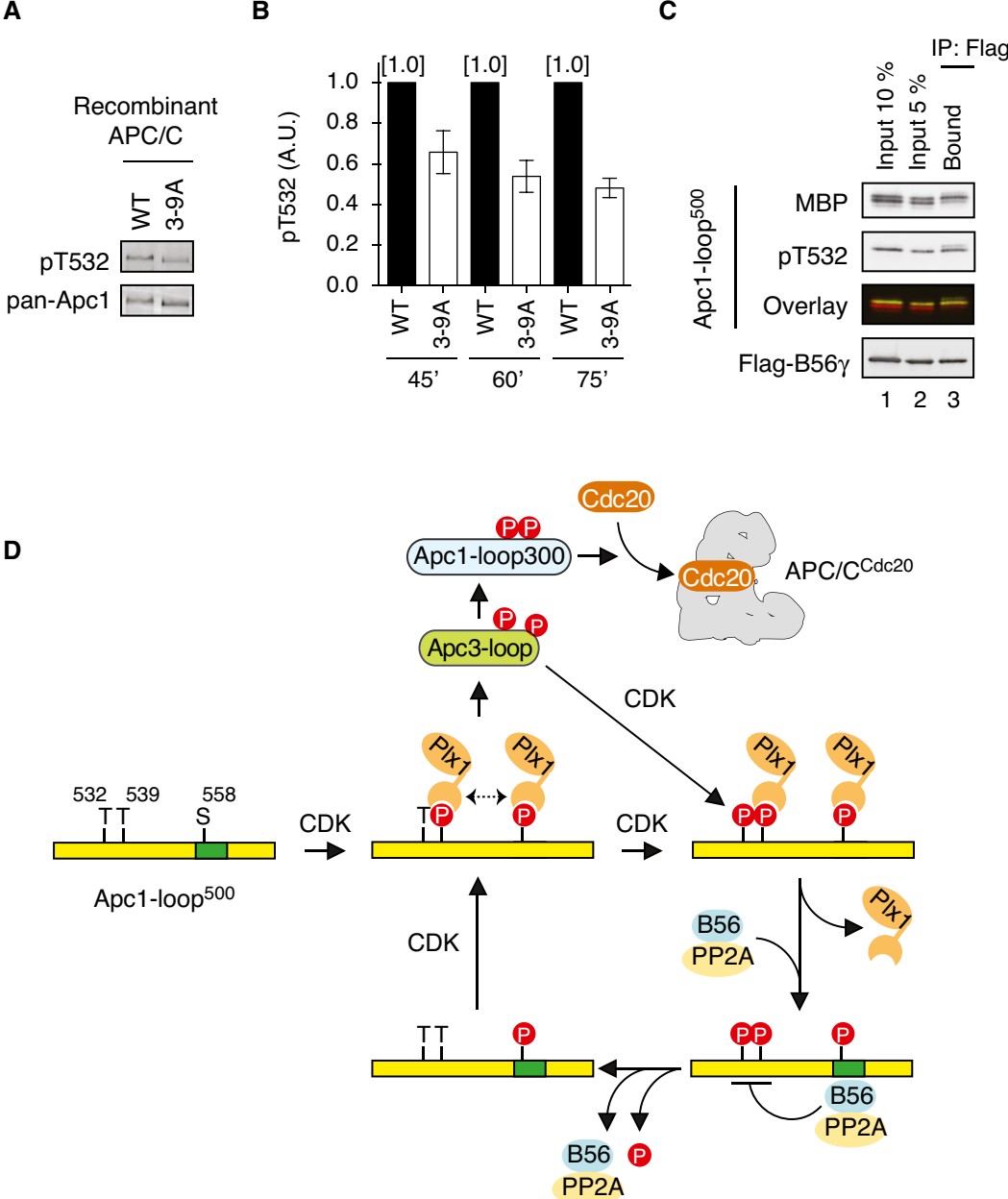

**Figure 7. A model: Plx1 and PP2A-B56 create feedback loop.**

A   The mutations in CDK sites on Apc3-loop reduces phosphorylation of T532 on Apc1. The purified recombinant WT or a mutant APC/C with mutations at nine CDK sites in Apc3-loop (3–9A) was incubated with APC/C-depleted (ΔAPC) anaphase extracts supplemented with non-degradable cyclin B at 23°C for 60 min. The APC/C was recovered with Apc3 monoclonal antibody (AF3.1) beads, and the phosphorylation status of T532 in Apc1 was analysed by SDS–PAGE and immunoblotting with phospho-specific (pT532) or Apc1 (pan-Apc1) antibodies.

B   Similar to (A), the experiments were performed at three time points, 45, 60 or 75 min in anaphase extracts. Quantification of phosphorylation levels of T532 is shown. The intensities of pT532 signals in WT at each time point (45, 60 or 75 min) were arbitrarily set to 1.0. The level of T532-phosphorylation was normalised to pan-Apc1. Error bars, SEM from three independent experiments.

C   PP2A-B56 preferentially binds to T532-phoshorylated Apc1-loop$^{500}$. Flag-tagged B56γ was incubated with the MBP-fused Apc1-loop$^{500}$ phosphorylated by Cdk2/cyclin A. The bound proteins were recovered by anti-flag affinity M2 beads, separated by SDS–PAGE and detected by immunoblotting with phospho-specific (pT532), MBP or Flag antibody. The signals of a phospho-specific antibody (pT532) and MBP were pseudo-coloured (green; pT532, red; MBP) and overlaid (Overlay).

D   A feedback control of APC/C by Plx1 and PP2A-B56 on Apc1-loop$^{500}$. CDK phosphorylates Apc1-loop$^{500}$ at T539 and S558 and promotes Plx1 binding onto Apc1-loop$^{500}$ via its PBD. Two PBD interactions, possibly dimerisation of Plx1, may stabilise the complex between Plx1 and Apc1-loop$^{500}$. Plx1 binding blocks PP2A-B56 recruitment to Apc1-loop$^{500}$ and stimulates Apc3-loop phosphorylation, which may induce Cdk1-CyclinB-p9/Cks2 recruitment and subsequent Apc1-loop$^{300}$ phosphorylation. Cdk1 recruited onto phosphorylated Apc3-loop promotes phosphorylation of T532, a residue important for the recruitment of PP2A-B56 on Apc1-loop$^{500}$. PP2A-B56 recruitment promotes dissociation of Plx1 by competitive binding and by dephosphorylation of T532 and T539, which diminishes Plx1-mediated phosphorylation of Apc3-loop. PP2A-B56 may quickly dissociate from Apc1-loop$^{500}$ once T532 is dephosphorylated, but Apc1-loop$^{500}$ is phosphorylated again by Cdk1 while Cdk1 is active. The dynamic phosphorylation–dephosphorylation cycle may continue until Cdk1 is inactivated, allowing swift response of the APC/C during the cell cycle.

appeared as two bands (Fig 7C lanes 1 and 2; MBP) and T532 phospho-specific antibodies (pT532) recognised the upper band exclusively (Fig 7C lanes 1 and 2; pT532 and overlay). When phosphorylated MBP-Apc1-loop$^{500}$ was incubated with Flag-B56γ, the upper band was selectively co-purified with Flag-B56γ (Fig 7C lane 3; pT532 and overlay) and this was dependent upon a B56-binding site present on Apc1-loop$^{500}$ (Appendix Fig S8A and C). It should be noted that S558 was fully phosphorylated under these conditions (Appendix Fig S8B lane 1; pS558 and overlay and Appendix Fig S8C). These results suggest that B56γ subunit preferentially binds to T532-phoshorylated Apc1-loop$^{500}$. Altogether, the data presented here show that Apc1-loop$^{500}$ is an important element in the control of the dynamic activity of the APC/C (Fig 7D, see Discussion).

## Discussion

A multi-subunit E3 ubiquitin ligase, the APC/C, is, alongside Cdk1, a key enzyme controlling entry and exit from mitosis. Studying the molecular mechanism of dynamic regulation of the APC/C is enormously challenging. Using a functional biochemistry pipeline with reconstituted APC/Cs, we recently demonstrated how Cdk1 activates the APC/C to initiate anaphase. This is one of the first glimpses of dynamic regulation of the APC/C (Fujimitsu *et al*, 2016; Qiao *et al*, 2016; Zhang *et al*, 2016a). Here, we show how an interplay between polo-like kinase (Plx1) and PP2A-B56 phosphatase plays a role in dynamic regulation of the APC/C. Plx1 and PP2A have been reported to control the APC/C (Patra & Dunphy, 1998; Rudner & Murray, 2000; Golan *et al*, 2002; Kraft *et al*, 2003; Labit *et al*, 2012; Hein *et al*, 2017; Lee *et al*, 2017; Fujimitsu & Yamano, 2020) but the molecular mechanism(s) of action are mostly unknown. Our study demonstrates that both Plx1 kinase and PP2A phosphatase exploit a flexible loop domain of Apc1, Apc1-loop$^{500}$, as an APC/C docking site but they mutually inhibit recruitment of the other. Intriguingly, recruitment of Plx1 and PP2A-B56 is determined by the phosphorylation of CDK sites (T532, T539, S558) on Apc1-loop$^{500}$, which differentially take place. According to prerequisite phosphorylation sites for Plx1 and PP2A-B56 recruitment, Plx1 is recruited to Apc1-loop$^{500}$ prior to PP2A-B56 and promotes APC/C-Cdc20 complex formation. Thus, we propose that in conjunction with Cdk1 activity, Apc1-loop$^{500}$ might function as a feedback control hub, fine-tuning optimal APC/C activity in mitosis (Fig 7D). In addition, Apc3-loop, another physically distant flexible loop from Apc1-loop$^{500}$, participates in this control, in which Cdk1 bound promotes phosphorylation of T532 and subsequently recruits PP2A-B56 to replace Plx1 on Apc1-loop$^{500}$, preventing over-activation of the APC/C (Fig 7D). Thus, our study highlights that the crosstalk of remote flexible loop domains in the APC/C is vital to coordinate APC/C activity with Cdk1 activity, allowing dynamic regulation of the APC/C.

Our study also reveals that Plx1 directly binds, via its PBD, to Apc1-loop$^{500}$ phosphorylated by CDK. Intriguingly, there are two tandem copies of the PBMs (PBM1 and PBM2) on a short 69 residue-Apc1-loop$^{500}$, which are highly conserved through vertebrates. Plx1 relies on PBM2 (STPAK), not PBM1 (STPLE), which matches to the genuine CDK consensus sequence [pS/pT]-P-X-[K/R]. In mitosis, T539, a requisite phosphorylation site for Plx1 binding in PBM2, is phosphorylated earlier than T532, a requisite

phosphorylation site for PP2A-B56 binding, in PBM1 (Fig 5). This observation is supported by our result that the impact of Plx1 recruitment is more prominent in early stage mitosis after activation of Cdk1 in *Xenopus* egg extracts. It may be that Plx1 on Apc1-loop$^{500}$ bolsters the phosphorylation of APC/C in early mitosis where Cdk1 activity is not high. It is reported that PP2A-B56 has a preference for dephosphorylation site which is dependent on surrounding sequences of the site and distance from B56-binding site (Kruse *et al*, 2020). The preferences of PP2A-B56 may contribute to the ordered phosphorylation of these CDK sites on Apc1-loop$^{500}$. In addition, Plx1 may have other docking sites in the APC/C other than Apc1-loop$^{500}$, and future studies to determine whether and how Plx1 is recruited to the other sites during control of the APC/C will be important, as Plx1 is clearly a key regulator during the cell cycle together with Cdk1.

PP2A-B56 recruitment onto Apc1-loop$^{500}$ is needed for the control of Plx1 dissociation through dephosphorylation of T539. This function of PP2A-B56 is required for precise activity control of the APC/C during mitosis as well as swift inactivation of the APC/C after Cdk1 inactivation (Fig 6). PP2A-B56 is probably recruited to Apc1-loop$^{500}$ only when Cdk1 activity is sufficiently high and Plx1 initiates Apc3 phosphorylation in order to avoid futile cycles of APC/C phospho-regulation. As S558 is more or less constantly phosphorylated in mitosis, a switch between Plx1-bound form and PP2A-B56-bound form is determined by the phosphorylation status of T539 and T532, respectively. PP2A-B56 preferentially dephosphorylates T539 and T532 (T539 > T532) over S558 (Fig 4B), which suggests that PP2A-B56 can dynamically interact with Apc1-loop$^{500}$ until Cdk1 is inactivated. Therefore, PP2A-B56 is a crucial element for dynamic regulation of the APC/C in mitosis (Fig 7D), allowing a swift response after Cdk1 inactivation. It is worth investigating whether PP2A-B56-driven dynamic regulation is common in phospho-regulation in mitosis. It has been recently reported that PP2A-B56 antagonises Plk1-loading onto kinetochore and that PP2A-B56-mediated removal of Plk1 from kinetochore is important for silencing of the spindle assemble checkpoint (Cordeiro *et al*, 2020). Thus, we surmise that the mode of dynamic regulation coordinated by Cdk1-Plx1/Plk1 and PP2A-B56 is a common molecular mechanism for a number of mitotic regulators or macromolecular complexes.

Short linear motifs (SLiMs) play a vital role in site-specific binding of proteins. Plx1 and B56 are reported to bind to the polo-binding motifs "S-[pS/pT]-[P/X]" (PBMs) and B56-binding motif (LxxIxE; Leu and Ile can be replaced with a hydrophobic residue), respectively (Elia *et al*, 2003a; Hertz *et al*, 2016; Wang *et al*, 2016; Wu *et al*, 2017). Our results demonstrate that along with these SLiMs, Plx1 and B56 need an additional site to acquire high affinities for Apc1-loop$^{500}$. As for Plx1, it is the sequence around S558 (Leu-Ser-Pro), which is slightly similar to a genuine PBM (Ser-pSer-Pro). In contrast, as for PP2A-B56, it is the sequence around the PBM1 including T532. These findings may perhaps be explained by either (i) an affinity model in which additional site contact increases its affinity for robust docking or (ii) more than one molecule binds to these sites for cooperative docking. The well-known PBD consensus, S-pS/pT-P, has been determined using short 16-residue peptides containing a single PBM (Elia *et al*, 2003a), yet two molecules of Plk1-PBD have been shown to bind to a peptide containing two PBMs with a higher affinity than a peptide containing a single PBM

(Jia *et al*, 2015). Also, peptide-bound Plk1-PBD can form a dimer through a hydrophobic patch in PBD (Zhu *et al*, 2016). Although the LSPV encompassing S558 is not a perfect PBM, it is likely that S558 directly interacts with PBD since S558A and L557A/V560A mutations reduce the binding of Plx1 (or Plx1-PBD) without affecting phosphorylation of T539 (Figs 3B and EV4). We thus favour the idea that two molecules of Plx1 bind to Apc1-loop$^{500}$ through T539 and S558 and the interaction between the two molecules stabilises the trimeric complex. This idea is supported by the result showing that Apc1-loop$^{500}$ can bind more than one molecule of Plx1-PBD (Fig EV1). However, we cannot exclude the possibility that a single molecule of Plx1-PBD can recognise both T539 and S558 residues in an as-yet unknown mechanism. The finding of the additional element for Plx1 binding might provide a new insight into a mechanism underlying site-specific binding of Plx1 and its homologue such as human Plk1. It is worth noting that Cdk1 and Plk1 have just been reported to promote kinetochore recruitment and dimerisation of Plk1 on Bub1 and CENP-U (Singh *et al*, 2021).

Although we find that PP2A-B56 recruitment relies on phosphorylation of T532, at the moment, we do not know the underlying mechanism for recognition or binding. In a crystal structure of the B56 bound to a peptide containing B56-binding motif, the motif is recognised in a concave surface in B56 HEAT repeat domain (Hertz *et al*, 2016; Wang *et al*, 2016). The HEAT repeat domain has several surfaces that could interact with an unstructured segment. In fact, a recent study has shown that a conserved acidic patch in the B56α HEAT repeat domain can interact with a basic charged rich region (2–4 residues) of substrate protein (Wang *et al*, 2020). We therefore speculate that an alternative surface on B56 HEAT repeat may interact with T532. The presence of multiple binding sites may help B56 selectively recognise substrate. It is possible that this feature is crucial for phosphatases in general as PP1 phosphatase has three modules for selective binding to PP1-interacting proteins including substrates (O'Connell *et al*, 2012; Peti *et al*, 2013; Kumar *et al*, 2018). Our findings suggest that multiple binding motifs on a flexible loop domain create an ingenious interplay of binding proteins through modification and even sharing of binding sequences, by which subunit–subunit communications and/or as a whole responsive-mode regulation of the APC/C can be accomplished.

We recently proposed a model, based on the results of analysis of the mutant APC/C (1-L557A/V560A and Δ11), that the PP2A-B56 associated with Apc1-loop$^{500}$ may dephosphorylate the N-terminal inhibitory domain of Cdc20 for APC/C-Cdc20 complex formation (Fujimitsu & Yamano, 2020). Unlike WT Cdc20, a non-phosphorylatable Cdc20 (Cdc20-5A) became insensitive to the mutations (1-L557A/V560A or Δ11) and Cdc20-5A was able to bind equally to both WT APC/C and mutant APC/Cs harbouring B56-binding site mutations. As this study reveals, the mutations severely inhibit both Plx1 and PP2A-B56 recruitment, so further investigation will be required for the detailed mechanisms of how PP2A-B56 associated with Apc1-loop$^{500}$ timely regulates APC/C-Cdc20 activity.

In summary, this study reveals how Cdk1 and Plx1 promote APC/C recruitment of Plx1 on Apc1-loop$^{500}$ and initiate phosphorylation of Apc3-loop, which in turn promotes phosphorylation of T532 responsible for PP2A-B56 recruitment, leading to dissociation of Plx1. Phospho-regulation is no doubt a key element in the APC/C system control, so this feedback control most likely helps to control APC/C activity not only during mitotic progression but also in response to environmental cues. Yet, we cannot exclude the possibility that Apc1-loop$^{500}$ could regulate phosphorylation of other APC/C subunits and binding partners, and vice versa. We are still far from understanding how the dynamic regulation of the APC/C is achieved with other mitotic kinases and phosphatases as well as cellular singling networks such as the spindle assembly checkpoint. Future work is definitely required to address the precise molecular basis of this regulation.

# Materials and Methods

### Antibodies and plasmid

Antibodies used are as follows: anti-Apc1 (RbAb 4853, 1:100), Apc3/Cdc27 (1:200; BD Transduction Laboratories), Apc5 (RbAb 3445, 1:500), Cdc20 (MAb BA8, 1:50), maltose-binding protein (MAb R29, 1:500), Plx1 (RbAb a gift from Dr Dunphy, 1:400) and Flag (1:1,000; SIGMA F-1804). Anti-Apc3 monoclonal antibody (MAb), AF3.1, was used for immunoprecipitation or immunodepletion of APC/C. Apc1 phosphopeptide antibodies (anti-pT532, 1:400; anti-pT539 1:50; anti-pS558 1:200) were raised in rabbits against LPH-conjugated phosphopeptide pT532: CMPRPSpTPLESI; pT539: CLESISpTPAKPL; and pS558: CEAGMLpSPVPEL (BioGenes, Germany). Antibodies were affinity-purified using a phosphopeptide column prepared with SulfoLink Kit (Pierce). After elution with ImmunoPure Gentle Ag/Ab Elution Buffer (Pierce), antibodies were dialysed against TBS and non-phospho-specific antibodies were affinity-depleted by passing through a column cross-linked with the corresponding non-phosphopeptide. The eluted phospho-specific antibodies were then enriched by dialysis against TBS containing 50% glycerol. Plasmids for the expression of *X. laevis* PP2A regulatory B subunits are gifts from Dr N. Sagata (Isoda *et al*, 2011).

### Preparation of *Xenopus* egg cell-free extracts

Meiotic metaphase II-arrested (CSF) *X. laevis* egg extracts were prepared as described (Murray, 1991). To prepare interphase extracts, CSF extracts were incubated at 23°C for 1.5 h in the presence of 0.4 mM CaCl$_2$ and 10 µg/ml cycloheximide, a protein synthesis inhibitor. Anaphase extracts were prepared by adding non-degradable GST-cyclin BΔ167 (a truncated form of human cyclin B lacking the N-terminal 167 amino acids) to interphase extracts and incubating for 30–60 min at 23°C (Fujimitsu *et al*, 2016). APC/C-depleted (ΔAPC) extracts were prepared as reported previously (Yamano *et al*, 2009).

### Expression and purification of recombinant APC/C

Expression and purification of recombinant APC/Cs were performed as described previously (Fujimitsu *et al*, 2016; Zhang *et al*, 2016b). Briefly, two baculoviruses carrying intact *Xenopus* APC/C genes and TEV-cleavable tandem Strep II-tag fused to Apc6 at C-terminus (Apc6-strep) were generated by MultiBac system (baculovirus 1: Apc1, Apc2, Apc10 and Apc11; baculovirus 2: Apc3, Apc4, Apc5, Apc6-strept, Apc7, Apc8, Apc12, Apc13, Apc15 and Apc16). Mutant APC/Cs were generated by PCR-based mutagenesis, and mutation sites were confirmed by sequencing. To express APC/C complex,

High Five insect cells (Invitrogen) at a cell density of $1.5 \times 10^6$ were co-infected with the two recombinant baculoviruses at an MOI (multiplicity of infection) of 1 for each virus and incubated at 27°C for 48 h with shaking (150 rev/min). The cells were harvested, frozen in liquid nitrogen and stored at −80°C. The recombinant APC/Cs were purified with Strep-Tactin beads and further affinity-purified by Dynabeads Protein A conjugated to anti-Apc3 monoclonal antibody (MAb AF3.1). The APC/Cs bound to beads were flash-frozen and stored at −80°C.

**Purification of recombinant proteins**

Expression and purification of Apc1-loop$^{500}$ (*Xenopus tropicalis* Apc1: 515–584) fused with PreScission protease-cleavable maltose-binding protein (MBP) at the N-terminus and a TEV-cleavable 6xHis at the C-terminus were performed as described previously (Fujimitsu & Yamano, 2020). The Apc3-loop, spanning amino acids 184–451 of *X. laevis* Apc3, was fused with a PreScission protease-cleavable 3xFlag-tag at the N-terminus and a TEV-cleavable 6xHis at the C-terminus and subsequently subcloned into pET vector. The PBD of Plx1 (Plx1-PBD), spanning amino acids 362–598 of *X. laevis* Plx1, was fused with a PreScission protease-cleavable 3xFlag-tag at the N-terminus and a TEV-cleavable 6xHis at the C-terminus and subsequently subcloned into pET vector. The mutants were generated by PCR-based mutagenesis. The resultant plasmids were introduced into BL21-CodonPlus (DE3) and the fusion proteins were expressed at 37°C for 1 h in the presence of 1 mM IPTG for Apc3-loop or 18°C overnight in the presence of 0.1 mM IPTG for Plx1-PBD. The cells were lysed by 0.3 mg/ml lysozyme and sonicated in lysis buffer (20 mM HEPES-NaOH pH 7.9, 500 mM NaCl, 5 mM EGTA, 10 μg/ml leupeptin, 10 μg/ml pepstatin A, 10 μg/ml chymostatin, 0.1% Triton X-100 and 10 mM imidazole). The proteins were purified from clarified lysate using Ni-NTA agarose beads (Qiagen). Recombinant p27$^{Kip1}$ protein tagged with 6xHis was expressed in BL21-CodonPlus (DE3) and purified using Ni-NTA agarose beads (Qiagen). To make Flag-tagged ubiquitin-fused lysine-less cyclin B (Ub-K0-cyclin B), the N-terminal 70-residue fragment of fission yeast cyclin B (N70) with no lysine residue was fused with a PreScission protease-cleavable Flag-tagged ubiquitin at the N-terminus and a TEV-cleavable 6xHis at the C-terminus, subcloned into pET vector and then expressed and purified using Ni-NTA agarose beads (Qiagen). Flag-tagged *X. laevis* B56γ was purified as described previously (Fujimitsu & Yamano, 2020).

**The Apc1-loop$^{500}$ binding assays**

For the binding assay of Apc1-loop$^{500}$ to endogenous Plx1, the purified MBP-fused Apc1-loop$^{500}$ fragment proteins were bound to amylose beads (New England Biolabs) by incubating at 4°C for 0.5–1 h. Beads were washed with Tris-NaCl buffer [20 mM Tris–HCl pH 8.0 and 200 mM NaCl] containing 0.01% NP-40, XB$^{CSF}$ buffer [10 mM HEPES-KOH pH 7.8, 50 mM sucrose, 100 mM KCl, 2 mM MgCl$_2$ and 5 mM EGTA] containing 0.01% NP-40 and XB$^{CSF}$ buffer. The resultant beads were incubated in interphase extracts in the presence or absence of non-degradable cyclin BΔ167 at 23°C for 60 min, separated from extracts on Micro Bio-Spin columns (Bio-Rad) and washed once with XB$^{CSF}$ buffer and then twice with XB$^{CSF}$ buffer containing 0.01% NP-40. The bound proteins were eluted

with SDS sample buffer and analysed by SDS–PAGE and immunoblotting. The binding assay of Apc1-loop$^{500}$ to Plx1-PBD was performed above except that Plx1-PBD was added after incubation with non-degradable cyclin B for 60 min and further incubated for 15 min. To see the impact of Cdk-dependent Apc1-loop$^{500}$ phosphorylation on Plx1-PBD binding, the MBP-fused Apc1-loop$^{500}$ fragment protein was incubated in the presence or absence of Cdk2/cyclin A in buffer (20 mM HEPES-KOH pH 7.8, 10 mM MgCl$_2$, 15 mM KCl, 1 mM EGTA, 1 mM ATP and 0.01% NP-40) at 30°C for 2hr and was bound to amylose beads (New England Biolabs) by incubating at 4°C for 0.5–1 h in Tris-NaCl buffer containing 0.01% NP-40. The beads were washed by Tris-NaCl buffer containing 0.01% NP-40 and XB$^{CSF}$ buffer containing 0.01% NP-40 without EGTA, resuspended into XB$^{CSF}$ containing 0.01% NP-40 without EGTA, incubated with Plx1-PBD at 4°C for 10 min, and then separated on Micro Bio-Spin columns (Bio-Rad), washed once with XB$^{CSF}$ buffer without EGTA and then twice with XB$^{CSF}$ buffer containing 0.01% NP-40 without EGTA. The bound proteins were eluted with SDS sample buffer and analysed by SDS–PAGE and Coomassie blue staining. The binding assay using MBP-fused Apc1-loop$^{500}$ fragments and [$^{35}$S]-labelled B56γ was performed as described previously (Fujimitsu & Yamano, 2020). To isolate B56-bound Apc1-loop$^{500}$ (Fig 7C and Appendix Fig S8), MBP-fused Apc1-loop$^{500}$ fragment proteins (10 μg) were first phosphorylated by Cdk2/cyclin A in 25 μl of buffer (20 mM HEPES-KOH pH 7.8, 10 mM MgCl$_2$, 15 mM KCl, 1 mM EGTA, 1 mM ATP and 0.01% NP-40) at 30°C for 2 h and then mixed with purified Flag-B56γ (5 μg) bound to anti-Flag affinity M2 beads (Sigma) in buffer (20 mM HEPES-KOH pH 7.8, 10 mM MgCl$_2$, 100 mM KCl, 1 mM EGTA, 1 mM ATP and 0.01% NP-40) at 4°C for 15 min. The beads were separated on Micro Bio-Spin columns (Bio-Rad) and washed once with XB$^{CSF}$ buffer and then twice with XB$^{CSF}$ buffer containing 0.01% NP-40. The bound proteins were eluted with SDS sample buffer and analysed by SDS–PAGE and immunoblotting.

**Immunoprecipitation of APC/C**

The APC/C was immunoprecipitated using Apc3 MAb (AF3.1) immobilised Dynabeads Protein A beads. The bound proteins were washed twice with XB$^{CSF}$_300 [XB$^{CSF}$ containing 300 mM KCl and 0.01% NP-40], eluted with SDS sample buffer and analysed by SDS–PAGE and immunoblotting.

**Cell-free destruction assay**

Destruction assays were performed as described previously (Fujimitsu *et al*, 2016). Substrates were labelled with [$^{35}$S]methionine (Hartmann Analytic, UK) in a coupled *in vitro* transcription–translation system (Promega, UK), and destruction assays were carried out using *Xenopus* egg cell-free extracts (anaphase or interphase extracts). The samples were taken at the indicated time points and analysed by SDS–PAGE and autoradiography. The images were analysed using ImageJ (NIH, USA).

**Ubiquitin assay**

Recombinant APC/Cs were incubated with *Xenopus* egg cell-free extracts in the presence of non-degradable cyclin B at 23°C for

45 min or 60 min and immunoprecipitated using Apc3 MAb (AF3.1)-immobilised Dynabeads Protein A. The beads were washed twice with $XB^{CSF}$_300 and once with Ub buffer (20 mM Tris–HCl pH 7.5, 100 mM KCl, 2.5 mM MgCl$_2$ and 0.3 mM DTT) containing 0.01% NP-40. The resultant APC/C-Cdc20 complexes were incubated at 23°C in 20 μl of Ub buffer containing 2 mM ATP, 50 ng/μl E1, 6 ng/μl Ube2S, 750 ng/μl methylated ubiquitin (MeUb) and 5 ng/μl Flag-ubiquitin-fused lysine-less N-terminal 70-residues of cyclin B (referred to as Ub-K0-cyclin B) as a substrate. The addition of MeUb prevents polyubiquitin chain formation, and a monoubiquitylated substrate facilitates quantification of APC/C activity. The reactions were stopped at the indicated time points with SDS sample buffer and analysed by SDS–PAGE and immunoblotting.

**Plx1 kinase assay**

The purified *X. laevis* Apc3-loop fragment proteins (80 μg/ml) were incubated in the $XB^{CSF}$ buffer containing 0.01% NP-40 and 4 μg/ml of purified Plx1 in the presence of 2 mM ATP or 100 μM [$^{32}$P]ATP at 23°C. The samples (2 μl) were taken at indicated times and analysed using SDS–PAGE, Coomassie blue staining or autoradiography.

**Apc1-loop$^{500}$ dephosphorylation assay and Plx1-PBD dissociation assay**

For dephosphorylation assay, the purified MBP-fused Apc1-loop$^{500}$ fragment proteins were bound to amylose beads (New England Biolabs) by incubating at 4°C for 0.5–1 h. Beads were washed with Tris-NaCl buffer containing 0.01% NP-40, $XB^{CSF}$ buffer containing 0.01% NP-40 and $XB^{CSF}$ buffer. The resultant beads were incubated in interphase extracts in the presence of non-degradable cyclin BΔ167 at 23°C for 60 min, washed twice with $XB^{CSF}$ buffer containing 0.01% NP-40 without EGTA, resuspended into $XB^{CSF}$ buffer containing 0.01% NP-40 without EGTA and incubated with purified PP2A-B56γ at 23°C. After reaction, the beads were separated on Micro Bio-Spin columns (Bio-Rad) and washed once with $XB^{CSF}$ buffer and then twice with $XB^{CSF}$ buffer containing 0.01% NP-40. The bound proteins were eluted with SDS sample buffer and analysed by SDS–PAGE and immunoblotting. Dissociation assay of Plx1-PBD was performed as described above except that Plx1-PBD was added after incubation with cyclin BΔ167 for 60 min and further incubated for 15 min, and the bound proteins were analysed by SDS–PAGE and Coomassie brilliant blue (CBB) staining.

**Statistical analyses**

Statistical analyses were performed in GraphPad Prism v6.0. Quantification data are presented as the mean ± SEM from three independent experiments.

# Data availability

This study includes no data deposited in external repositories.

**Expanded View** for this article is available online.

## Acknowledgements

We thank Drs N. Sagata for plasmids expressing PP2A regulatory B subunits; W. Dunphy for anti-Plx1 antibody; H. Labit for *X. laevis* PP2A-B56c baculovirus; the staff at the UCL Biological Services Unit for taking care of the *Xenopus* colony at UCL; and M. Grimaldi and members of the Yamano laboratory for helpful discussions and critical reading of the manuscript. This work was supported by the Wellcome Trust (205150/Z/16/Z).

## Author contributions

KF and HY conceived and designed the project. FK conducted all the experiments under the supervision of HY. KF and HY wrote the manuscript.

## Conflict of interest

The authors declare that they have no conflict of interest.

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
