## [Review Process File · The EMBO Journal]

Dynamic regulation of mitotic ubiquitin ligase APC/C by coordinated Plx1 kinase and PP2A phosphatase action on a flexible Apc1 loop

Hiroyuki Yamano and Kazuyuki Fujimitsu
DOI: [10.15252/emboj.2020107516](https://doi.org/10.15252/emboj.2020107516)

Corresponding authors: Hiroyuki Yamano (h.yamano@ucl.ac.uk)

Review Timeline:

Submission Date:	14th Dec 20
Editorial Decision:	19th Jan 21
Revision Received:	3rd Jun 21
Editorial Decision:	28th Jun 21
Revision Received:	2nd Jul 21
Accepted:	6th Jul 21

Editor: Hartmut Vodermaier

Transaction Report:

Thank you for submitting your manuscript on Plx1 and PP2A coordination by an APC1 loop region to our editorial office. I have now received the reports of three expert referees, who further discussed their views with each other and with me in our cross-commenting forum. Based on the reports and on these discussion, we would be interested in considering this work further for The EMBO Journal, provided that you should be prepared and able to extend this work further towards increased functional relevance of the data.

As you will see from the comments included below, all reviewers clearly appreciate the importance of the topic and the experimental standards and quality of the presented data. However, the major limitation initially pointed out by referee 1, and subsequently conceded also by referees 2 and 3, is the current over-reliance on experiments using isolated MBP-fused loop peptides outside the context of APC/C, thereby jeopardizing the significance of any models for the mechanism of holo-APC/C regulation. In this light, we feel that it will be essential to extend the work further by using your recombinant APC/C system as well as functional assays in ubiquitination and mitotic exit, in order to make it a compelling candidate for a full EMBO Journal article. I realize that this may require considerable further time and effort, and would be happy to offer an extended revision deadline for achieving it. Alternatively, should you prefer to rather publish this work rapidly and without major extensions, resubmission/transfer to sister journals such as EMBO reports or Life Science Alliance would be options we might also discuss.

In case you decide to comprehensively revise the study for The EMBO Journal, please make sure to carefully answer also to the various more specific points raised in all three reports. Please address in particular possible alternative roles of the various docking motifs as suggested by the referees, and try to better reconcile the present work with the findings you reported in your recent EMBO reports publication.

Please do not hesitate to get back to me once you have considered the reports in depth together with your collaborators, I would be happy to talk about the comments and revision possibilities also directly with you via phone or video call.

Thank you again for the opportunity to consider this work for The EMBO Journal. I look forward to hearing from you in due time.

Referee #1:

In this manuscript the regulation of the APC/C-Cdc20 complex is investigated with a focus on a regulatory loop in APC1 (loop 500). The APC/C-Cdc20 complex is an important regulator of cell division and is heavily regulated by phosphorylation. Previous work have implicated both Cdk1 and Plk1 phosphorylations of APC/C as important activating mechanisms although the molecular details of are not fully understood. The Yamano lab and other labs have shown that phosphoregulation of APC1 loop 300 by Cdk1 is important because it controls the binding of Cdc20. APC1 loop 300 phosphorylation is mediated by binding of Cyclin B1-Cdk1-Cks to APC3. Less is known about how Plk1 phosphorylates the APC/C and also the role of protein phosphatases in reversing these phosphorylations.

Here the authors undertake a detailed analysis of the APC1 loop 500 that contains two putative Plk1 (Plx1 in *Xenopus*) docking sites (T532 and T539) as well as a binding site for PP2A-B56. The latter was recently investigated by the authors in an EMBO Reports paper where they argued that binding of PP2A-B56 to APC1 loop500 is required for Cdc20 dephosphorylation and binding/activation of APC/C. The authors establish that T539 is a binding site for Plx1 and that PP2A-B56 bound to the LxxIxE motif dephosphorylates T532 and T539 hereby antagonizing Plx1 binding to APC1. Binding of PP2A-B56 to APC1loop500 is dependent on T532 phosphorylation while T539 phosphorylation is not required. Furthermore, they show that the Cdk1 phosphorylation site S558 within the PP2A-B56 binding site is required for both PP2A-B56 binding and Plx1 binding and that there appear to be competition between PP2A-B56 and Plx1 for binding to APC1. The authors also establish that binding of Plx1 to APC1 loop500 contributes to APC3 phosphorylation and Cdc20 binding leading to premature activation of APC/C.

My overall reflections of this manuscript are the following:

The experiments are carefully controlled but in most instances the authors work with MBP-APC1 loop500 fragments rather than looking at binding and phosphorylation events in the context of the APC/C. One has to question how relevant this is and whether the results can be directly transferred to the APC/C complex. Furthermore, there is a limited testing of the proposed mechanisms and how this affects APC/C activity and mitotic exit (only one mutant is analyzed in 6C for Cyclin degradation and the effects are minor). More careful analysis of APC/C mutants in mitotic exit assays and in vitro ubiquitination assays is needed.

I am also not convinced that all aspects of APC1 loop500 is elucidated in this work. The T532 looks like a good fit for the Cks1 consensus but not the Plx1 consensus (due to P530 - McGrath 2013) but the authors do not investigate if CyclinB1-Cdk1-Cks1 docks here to mediate S558 phosphorylation which would be in my view the most straight forward explanation for why T532A blocks PP2A-B56 binding. Furthermore, the authors do not consider the role of S567 phosphorylation by Plx1 which would stimulate PP2A-B56 binding (very similar to the Plk1 docking at S620 in human BubR1 and phosphorylation of S676/T680 to drive PP2A-B56 recruitment). Thus, I feel more work is needed to dissect the regulation of APC1 loop500 also in light of the authors not showing strong biological relevance of their current findings. In the discussion I also think it is important that they make it clear that the results in Figure 6 is in direct contrast to the model they recently put forward in EMBO Reports as shown by the E562A mutant. The E562A appears to be a much cleaner separation of function mutant than the L557A/V560A mutant they used in EMBO Reports.

Given the limited analysis of biological relevance and still many aspects of APC1 loop500 regulation not fully clarified I do not find the manuscript mature enough for the broader audience of EMBO J.

Specific comments/suggestions:

- 1) Figure 5: Why do they not investigate the phosphorylation of APC1 in the context of the APC/C with their phosphoabs? Maybe the kinetics are very different
- 2) They use Cdc20 binding as a proxy for APC/C activity but without activity assays it is unclear if this is a proper measure.
- 3) Page 12: I think S558 is a poor substrate for PP2A-B56 because it is an SP site and is binding to PP2A-B56 rather than because it is a Ser (see Kruse et al 2020).
- 4) On Page 19: The authors speculate that S558 could be a PBD binding site. As the authors state in the introduction the PBD consensus is S-pS/pT-P and to my knowledge the serine before the phosphorylation site is absolutely required (Elia et al 2003).
- 5) The authors have some very powerful tools to fully elucidate Plx1 phosphorylation and PP2A-B56 regulation of APC/C-Cdc20 by combining their APC/C mutants with phosphoproteomics - I think these types of experiments would add a lot by providing a "global" view of APC/C phosphoregulation.
- 6) In the Abstract and in Figure 5 the authors write: "...premature activation APC/C activation and delays mitotic exit". Premature APC/C activation should result in premature mitotic exit
- 7) Better characterisation of phosphorylation abs - in particular lambda phosphatase treatment to establish if there is still some recognition after full dephosphorylation (in Fig 4 it is unclear if some of the bands remaining is due to antibodies also recognising the unphosphorylated form)

Jakob Nilsson

Referee #2:

In this manuscript by Fujimitsu and Yamano, the authors examine the intricate phosphor-regulation of the APC/C. The APC/C is an essential ubiquitin ligase for cell cycle control, and its activity must be tightly controlled. However, a detailed understanding of the dynamic interplay between kinases and phosphatases have remained largely elusive. Relevant to this manuscript, the coordinated action between the kinases Cdk1 and Plx1 and the phosphatase PP2A is explored using numerous constructs, pulldown assays, Xenopus egg extracts, and recombinant APC/C. Taken together, this paper provides significant insight into the site-specific phosphorylation of the APC/C and the competition between Plx1 and PP2A. Therefore, it is my opinion that this study is suitable for publication in The EMBO Journal and the authors should address the following points prior to publication.

- 1) While the manuscript nicely presents the site-specific recruitment and binding of Plx1 and PP2A onto the APC1-loop500, its involvement in the crosstalk relationship with the APC3 loop, responsible of CDC20 binding, is seemingly more tentative. For example, the Plx1 mutants in the APC1-loop500 seem to decrease APC3 phosphorylation (Figure 2A-B), but the follow-up experiment to validate this result is missing a negative control to show the specificity of the in vitro phosphorylation. Similarly, how the distant APC3-loop initiates the PP2A recruitment is based on an experiment using an indirect readout, phosphorylation status of T532. Additional experiments could help strengthen this aspect of the paper.
- 2) Due to the complex regulation of and by Plx1 and PP2a, it is understandably difficult to simply

convey many of the points in the manuscript. However, every effort should be made to provide clarity for the reader by combining experimental findings when possible. For example, the APC1 S558A substitution is used in Figure 1, but an explanation as to why this would impact Plx1 binding when it is found in the PP2A-B56-binding motif is not given until Figure 4.

Minor comments

- 1) The kinetics of the phosphorylation shown in Figure 5 is important to understand the sequence of phosphorylation events. However, it is performed on the MBP-APC1-loop500. Since these are phospho-site specific antibodies, examining the kinetics using the whole APC/C could strengthened the conclusions.
- 2) The mutually competitive binding mode of PP2A and Plx1 could have further implications in the phosphor-regulation of other APC/C subunits and binding partners. Perhaps these implications could be included in the Discussion.
- 3) On page 5, it reads "APC/C complex" when the first "C" stands for complex.

Referee #3:

The timely activation of kinases and phosphatases is essential for the orderly progression of the cell cycle. The Plk1 and Cdk1 kinases and the PP2A-B56 phosphatase have been previously implicated as regulators of a modular organization of docking and phosphorylation sites in many target proteins. A detailed description of these regulatory network is tedious, difficult, and often full of surprises and unexpected results. Nonetheless, these studies are important and welcome, as they will ultimately shed light on how the activity of essential cell cycle machinery is regulated in time.

This manuscript by Fujimitsu and Yamano is technically excellent. It reports a detailed analysis of one such regulatory network staged on loop regions of the Apc1 and Apc3 subunits of the anaphase promoting complex/cyclosome (APC/C), a ubiquitin ligase required for mitotic exit. Somehow, the study opens more questions than it answers, as explained below, but this is also an important realization that will advance future studies, and a merit of the study is that it seems to provide excellent reliable data.

Briefly, the authors analyzed the effects of various mutations in a module with a strictly conserved configuration of docking and phosphorylation sites within the Apc1-500 loop (so called to distinguish it from another Apc1 loop previously shown, also by the authors, to be regulated by phosphorylation and to be part of the same regulatory network). This module contains two CDK sites being construed to be also docking sites for PLK1 upon their phosphorylation, and a PP2A-B56gamma docking site (described in a previous recent study by the authors), also containing a potential CDK site. The authors demonstrate that perturbations of these motifs have "long distance" effects on the phosphorylation of Apc3 and also on the interaction of Cdc20 with the APC/C and its activation, and that this is reciprocal, because Apc3 phosphorylation status influences the phosphorylation of Apc1 on the 500 loop.

Collectively, I feel that this study has many merits and that it has the potential to be of interest to the wide audience of the EMBO Journal. The text is straightforward and the figures are usually self-explanatory. I would appreciate if the authors took note of the following comments and

suggestions.

-Abstract: "Stable Plx1 binding...delays mitotic exit" I guess that the authors mean accelerates, not delays? This is what I infer from Figure 6C.

-Page 7: "anaphase extracts" are introduced without further clarifications, and this will be confusing, as most readers will associate anaphase with the period when Cyclin B is degraded. The authors should clarify that these extracts are supplemented with non-degradable Cyclin B.

-Probably the most striking effect described in this study arises from mutating Ser558 to Ala. This mutant ablated not only the interaction of PP2A-B56 with the Apc1-500 loop, but also the interaction of Plx1 with the upstream sites T532 and T539. The authors argue that Ser558, being part of a "degenerate" PBD binding site, LSP, may contribute to stabilization of Plx1 binding through its PBD. I note that the early work of Mike Yaffe and colleagues (Elia et al. 2003) seems to exclude this because any ligand with side chains other than Ser, even Thr, on the residue preceding the phospho-site were unable to interact with the PBD. Could it rather be that the LSPV sequence encompassing S558 is a docking site for the Cks2/Cyclin/CDK complex? A possible expectation if the authors hypothesis were correct is that T532 and T539 continue to be phosphorylated when Ser558 is mutated. A possible expectation of the alternative hypothesis I propose here is that these sites are not phosphorylated (even when PP2A activity is inhibited). Testing this should be within easy reach with the tools available to the authors.

-A related issue is raised from a statement in the Discussion: If two molecules of Plx1 bind to Apc1-loop 500 through T539 and S558, as proposed by the authors, why doesn't the T532A mutant, which cannot bind PP2A (like E562A) have at least as much Plx1 bound as WT?

-On page 9, the authors demonstrate that Plx1 can phosphorylate Apc3-loop, which is interesting, even if ultimately the authors do not test the significance of this observation. Regardless, a question for this and other kinase assays is how the authors think of the specificity of these phosphorylation events. We see Plx1 can phosphorylate targets, but is this selective for Plx1 or would other mitotic kinases, under similar conditions, do the same? Can the authors comment on this point?

-On page 10 and page 13 (two instances) the authors refer to an E532A mutation when I think they really mean E562A in all three cases. Please check.

-Page 17: streamline sentence "In addition, another physically distant flexible loop from Apc1-loop, Apc3-loop, participates in the control...". Could rather read "In addition, Apc3-loop, another flexible loop physically distant from Apc1-loop, participates in the control..."

-Page 18: "Lue" should read "Leu"

-In general, many of the experiments here rely on the assumption that APC loop regions fused to MBP are reasonable proxies for their behavior on the APC/C. This is reasonable but the authors may elect to include a statement to warn readers that they are aware that this is not necessarily true.

RE: Manuscript EMBOJ-2020-107516
Point-by-Point response to reviewers' comments

We thank all reviewers for their insightful and positive comments, which we feel have substantially improved the manuscript. Please find below our point-by-point response in blue text.

Referee #1:

*My overall reflections of this manuscript are the following:
The experiments are carefully controlled but in most instances the authors work with MBP-APC1 loop500 fragments rather than looking at binding and phosphorylation events in the context of the APC/C. One has to question how relevant this is and whether the results can be directly transferred to the APC/C complex. Furthermore, there is a limited testing of the proposed mechanisms and how this affects APC/C activity and mitotic exit (only one mutant is analyzed in 6C for Cyclin degradation and the effects are minor). More careful analysis of APC/C mutants in mitotic exit assays and in vitro ubiquitination assays is needed.
I am also not convinced that all aspects of APC1 loop500 is elucidated in this work. The T532 looks like a good fit for the Cks1 consensus but not the Plx1 consensus (due to P530 - McGrath 2013) but the authors do not investigate if CyclinB1-Cdk1-Cks1 docks here to mediate S558 phosphorylation which would be in my view the most straight forward explanation for why T532A blocks PP2A-B56 binding. Furthermore, the authors do not consider the role of S567 phosphorylation by Plx1 which would stimulate PP2A-B56 binding (very similar to the Plk1 docking at S620 in human BubR1 and phosphorylation of S676/T680 to drive PP2A-B56 recruitment). Thus, I feel more work is needed to dissect the regulation of APC1 loop500 also in light of the authors not showing strong biological relevance of their current findings. In the discussion I also think it is important that they make it clear that the results in Figure 6 is in direct contrast to the model they recently put forward in EMBO Reports as shown by the E562A mutant. The E562A appears to be a much cleaner separation of function mutant than the L557A/V560A mutant they used in EMBO Reports.
Given the limited analysis of biological relevance and still many aspects of APC1 loop500 regulation not fully clarified I do not find the manuscript mature enough for the broader audience of EMBO J.*

As suggested, we have used the whole APC/C and analysed the activities and regulation. The new results support our original model that Apc1-loop⁵⁰⁰ is important for dynamic phosphoregulation of APC/C. New data are presented in Figs 2C, 5C and EV5 and Appendix Figs S2, S3, S4 and S8.

It was surprising even for us that T532A mutation inhibits the binding of PP2A-B56 to Apc1-loop⁵⁰⁰. We also thought that T532 could promote the phosphorylation of S558 in the early stage of this study. However, using phospho-site specific Ab, pS558, we have confirmed that the phosphorylation status of S558 is almost the same between WT and T532A mutation in anaphase extracts where CyclinB-Cdk1-Cks1 is present (See Fig EV4A). Thus, it is unlikely that Cyclin B1-Cdk1-Cks1 docks on pT532 and subsequently phosphorylates S558.

We have also investigated the phosphorylation status of T532 on Apc1-loop⁵⁰⁰ when B56γ is bound and found that T532 is heavily phosphorylated on B56 bound Apc1-loop⁵⁰⁰. This suggests that B56 specifically binds Apc1-loop⁵⁰⁰ in which T532 is phosphorylated,

indicating that T532 is required for B56 loading onto Apc1-loop⁵⁰⁰. This new result is presented in Fig 7C and Appendix Fig S9.

We appreciate the reviewer's suggestion on S566/T567, so we examined the impact of the mutation of S566A/T567A and found that these mutations had no impact on either Plx1-binding or B56-binding. The new result is presented in Appendix Fig S6.

Our data in EMBO Reports suggest the importance of PP2A-B56 bound to Apc1-loop⁵⁰⁰ for Cdc20 dephosphorylation and Cdc20-APC/C formation. This is also supported by the result with a non-phosphorylatable mutant Cdc20 that can efficiently bind the APC/C even when PP2A-B56 binding is impeded (rescue the phenotype of L5567A/V560A mutant). We still believe this is essentially correct; PP2A-B56 is important for APC/C activation. However, the underlying mechanism might not be simply that the sole target of PP2A on Apc1-loop⁵⁰⁰ is Cdc20. In fact, in our model in EMBO Report (Fig 4F), we have drawn a dotted arrow from B56 towards the APC/C. In this manuscript, our results show that PP2A-B56 on Apc1-loop⁵⁰⁰ is important for dynamic regulation of the APC/C although we do not fully understand the underlying mechanism as shown in the Discussion. Yet, we have provided significant insight into the site-specific phosphorylation of the APC/C as well as the interplay between Plx1 and PP2A-B56 on Apc1-loop⁵⁰⁰ for control of the APC/C. We have tried to isolate Apc1-loop⁵⁰⁰ mutants with a clearer separation of function (either Plx1 or B56-binding) by alanine scanning mutagenesis but this has proved much more difficult than expected and without success. As the reviewer appreciates, the E562A appears to be clearer than L5567A/V560A mutant, but it seems to behave as a gain-of-function mutant compared with WT and therefore is not straightforward. Stabler Plx1 binding apparently drives APC/C activation more dominantly than WT and thus the impact from PP2A-B56 binding deficiency seems occluded. Still, the E562A mutant is useful, shedding light on Plx1 and the dynamic regulation of APC/C.

Specific comments/suggestions:

1) Figure 5: Why do they not investigate the phosphorylation of APC1 in the context of the APC/C with their phosphoabs? Maybe the kinetics are very different

As suggested, we investigated the phosphorylation of Apc1 on the APC/C in mitosis. As shown in the new Fig. xx, the result is consistent with the kinetics studied using the fragment MBP-Apc1-loop⁵⁰⁰. The new result is presented in Fig 5C and EV5.

2) They use Cdc20 binding as a proxy for APC/C activity but without activity assays it is unclear if this is a proper measure.

As suggested, we have investigated the activity of apoAPC/Cs with Apc1-loop⁵⁰⁰ mutations. Consistent with Cdc20 binding, S558A and T532A/T539A mutations compromised the APC/C-mediated ubiquitylation activity as well as APC/C-mediated cyclin destruction in anaphase. New data are presented in Fig 2C and Appendix Fig S2 and S3.

3) Page 12: I think S558 is a poor substrate for PP2A-B56 because it is an SP site and is binding to PP2A-B56 rather than because it is a Ser (see Kruse et al 2020).

As suggested, we have added.

4) On Page 19: The authors speculate that S558 could be a PBD binding site. As the authors state in the introduction the PBD consensus is S-pS/pT-P and to my knowledge the serine before the phosphorylation site is absolutely required (Elia et al 2003).

We appreciate the breakthrough discovery of PBD-phosphopeptide recognition by the Yaffe laboratory, yet in the binding assay Elia et al used short peptides (16 residues). In contrast, we have used 69-residue fragments that might contain not only the genuine PBMs but also several possible Plx1 binding or recognition sites. Several recent reports suggest that a tandem orientation of Plk1-binding sites stabilises Plk1-binding (Jia et al 2015, Singh et al 2020), likely through PBD-PBD interaction (Zhu et al 2016).

We have reconstituted PBD and Apc1-loop⁵⁰⁰ binding assay in the presence of Cdk2/cyclin A. New data are presented in Appendix Fig S1. S558A mutation reduced its binding to nearly 50% even though both T532 and T539 sites (the PBD consensus sites) are well phosphorylated (Fig EV4B). In the same assay (Appendix Fig S1), we could confirm our result (Fig. 1C) : 1) T539 phosphorylation is more important than T532 for Plx1 binding. 2) when both T532 and T539 are mutated (2A), Plx1 shows no binding to Apc1-loop⁵⁰⁰ even if S558 is phosphorylated. Thus, as appreciated in the field, the PBD consensus is vital for Plx1 binding. In addition, our result implies that phosphorylation of S558 plays a role for Plx1 binding towards the PBD-PBM interaction.

At the moment, we do not know the underlying mechanism, but we speculate that the orientation of the sites (T539 and S558) and/or the distance between them may be just appropriate for the formation of a hetero-trimer complex including Plx1 (two molecules) and Apc1-loop⁵⁰⁰. It is also possible that the requirement of Plx1-binding is slightly relaxed because of specific adjacent sequences or an as-yet unidentified mechanism. The detailed study of the mechanism underlying this cooperative binding is intriguing, but we believe that this is beyond the scope of this manuscript.

5) The authors have some very powerful tools to fully elucidate Plx1 phosphorylation and PP2A-B56 regulation of APC/C-Cdc20 by combining their APC/C mutants with phosphoproteomics - I think these types of experiments would add a lot by providing a "global" view of APC/C phosphoregulation.

We appreciate the reviewer's suggestion. It would be great if we could obtain a "global" view of APC/C phosphoregulation, but the work is clearly beyond the scope of this manuscript.

6) In the Abstract and in Figure 5 the authors write: "...premature activation APC/C activation and delays mitotic exit". Premature APC/C activation should result in premature mitotic exit

We apologise for the misleading expression. We have changed to "delays APC/C dephosphorylation during mitotic exit". According to Fig 6D, the mutant APC/C (1-E562A)

delays APC/C dephosphorylation because of stronger Plx1 binding which prevents efficient APC/C dephosphorylation during mitotic exit.

7) Better characterisation of phosphorylation abs - in particular lambda phosphatase treatment to establish if there is still some recognition after full dephosphorylation (in Fig 4 it is unclear if some of the bands remaining is due to antibodies also recognising the unphosphorylated form)

We show the site-specificity for phospho-specific antibodies in EV4. The phospho-specific antibodies do not recognise the site when it is not phosphorylated, demonstrated in Fig. 5A. At time 0, when Apc1-loop⁵⁰⁰ is not phosphorylated, there is no signal, and signals appear only after mitotic induction by adding non-degradable cyclin B, indicating that our phospho-antibodies specifically recognise phosphorylation of T532, T539 and S558.

Referee #2:

In this manuscript by Fujimitsu and Yamano, the authors examine the intricate phosphoregulation of the APC/C. The APC/C is an essential ubiquitin ligase for cell cycle control, and its activity must be tightly controlled. However, a detailed understanding of the dynamic interplay between kinases and phosphatases have remained largely elusive. Relevant to this manuscript, the coordinated action between the kinases Cdk1 and Plx1 and the phosphatase PP2A is explored using numerous constructs, pulldown assays, Xenopus egg extracts, and recombinant APC/C. Taken together, this paper provides significant insight into the site-specific phosphorylation of the APC/C and the competition between Plx1 and PP2A. Therefore, it is my opinion that this study is suitable for publication in The EMBO Journal and the authors should address the following points prior to publication.

We thank the reviewer for his/her positive comments on our study.

1) While the manuscript nicely presents the site-specific recruitment and binding of Plx1 and PP2A onto the APC1-loop500, its involvement in the crosstalk relationship with the APC3 loop, responsible of CDC20 binding, is seemingly more tentative. For example, the Plx1 mutants in the APC1-loop500 seem to decrease APC3 phosphorylation (Figure 2A-B), but the follow-up experiment to validate this result is missing a negative control to show the specificity of the in vitro phosphorylation. Similarly, how the distant APC3-loop initiates the PP2A recruitment is based on an experiment using an indirect readout, phosphorylation status of T532. Additional experiments could help strengthen this aspect of the paper.

In order to show the specificity of the in vitro phosphorylation by Plx1, alpha-casein, Cdc20 NTD (N159), MBP or Apc3-loop were incubated in the presence of ³²P-ATP and Plx1. With the exception of MBP, all substrates are phosphorylated by Plx1 in time-dependent manner, in particular, Apc3-loop being the most phosphorylated among the four proteins tested, suggesting good specificity towards Apc3-loop of Plx1. New data are presented in Fig 2E and Appendix Fig S5.

To investigate the relation between Plx1-binding and Apc3 phosphorylation, we have monitored the phosphorylation status of Apc3 within the APC/C in which Apc1-loop⁵⁰⁰ has mutations of T532A/T539A or S558A in anaphase. When Plx1 binding to Apc1-loop⁵⁰⁰ is blocked, Apc3 phosphorylation is delayed about 8~10 min, compared with WT. It may be clearer in Phos-tag gel: once Apc3 is highly phosphorylated, the band disappears, presumably due to hyperphosphorylation and phos-tag-mediated retardations, which occur at 50 min in WT but at 60min in T532A/T539A or S558A. New data are presented in Appendix Fig S4.

To study the distant regulation is not a simple task, but to investigate the importance of phosphorylation of T532 on the binding of B56 to Apc1-loop⁵⁰⁰, we have examined the phosphorylation state of T532 on the Apc1-loop⁵⁰⁰ bound to B56 γ . As mentioned in the comment to reviewer 1, B56 γ subunit specifically binds T532-phosphorylated Apc1-loop⁵⁰⁰ fragments even when S558 is similarly phosphorylated. This result supports the importance of Apc3-directed phosphorylation of T532 in recruitment of PP2A-B56 on Apc1-loop⁵⁰⁰. This new result is presented in Fig 7C and Appendix Fig S9.

2) Due to the complex regulation of and by Plx1 and PP2a, it is understandably difficult to simply convey many of the points in the manuscript. However, every effort should be made to provide clarity for the reader by combining experimental findings when possible. For example, the APC1 S558A substitution is used in Figure 1, but an explanation as to why this would impact Plx1 binding when it is found in the PP2A-B56-binding motif is not given until Figure 4.

We added a description in the text.

Minor comments

1) The kinetics of the phosphorylation shown in Figure 5 is important to understand the sequence of phosphorylation events. However, it is performed on the MBP-APC1-loop500. Since these are phospho-site specific antibodies, examining the kinetics using the whole APC/C could strengthened the conclusions.

This is the same as the point 1 (Reviewer 1). As suggested, we have used the whole APC/C and investigated the phosphorylation of Apc1. As shown in the new Fig 5C and EV5, the result is consistent with the kinetics studied using the fragment MBP-Apc1-loop⁵⁰⁰.

2) The mutually competitive binding mode of PP2A and Plx1 could have further implications in the phosphor-regulation of other APC/C subunits and binding partners. Perhaps these implications could be included in the Discussion.

As suggested, in the Discussion, we have included the possibility of the implications of the phospho-regulation of other APC/C subunits and binding partners.

3) On page 5, it reads "APC/C complex" when the first "C" stands for complex.

We corrected.

Referee #3:

The timely activation of kinases and phosphatases is essential for the orderly progression of the cell cycle. The Plx1 and Cdk1 kinases and the PP2A-B56 phosphatase have been previously implicated as regulators of a modular organization of docking and phosphorylation sites in many target proteins. A detailed description of these regulatory network is tedious, difficult, and often full of surprises and unexpected results. Nonetheless, these studies are important and welcome, as they will ultimately shed light on how the activity of essential cell cycle machinery is regulated in time.

This manuscript by Fujimitsu and Yamano is technically excellent. It reports a detailed analysis of one such regulatory network staged on loop regions of the Apc1 and Apc3 subunits of the anaphase promoting complex/cyclosome (APC/C), a ubiquitin ligase required for mitotic exit. Somehow, the study opens more questions than it answers, as explained below, but this is also an important realization that will advance future studies, and a merit of the study is that it seems to provide excellent reliable data.

Briefly, the authors analyzed the effects of various mutations in a module with a strictly conserved configuration of docking and phosphorylation sites within the Apc1-500 loop (so called to distinguish it from another Apc1 loop previously shown, also by the authors, to be regulated by phosphorylation and to be part of the same regulatory network). This module contains two CDK sites being construed to be also docking sites for PLK1 upon their phosphorylation, and a PP2A-B56 docking site (described in a previous recent study by the authors), also containing a potential CDK site. The authors demonstrate that perturbations of these motifs have "long distance" effects on the phosphorylation of Apc3 and also on the interaction of Cdc20 with the APC/C and its activation, and that this is reciprocal, because Apc3 phosphorylation status influences the phosphorylation of Apc1 on the 500 loop.

Collectively, I feel that this study has many merits and that it has the potential to be of interest to the wide audience of the EMBO Journal. The text is straightforward and the figures are usually self-explanatory. I would appreciate if the authors took note of the following comments and suggestions.

We thank the reviewer for his/her positive comments on our study.

-Abstract: "Stable Plx1 binding...delays mitotic exit" I guess that the authors mean accelerates, not delays? This is what I infer from Figure 6C.

We apologise for the misleading expression. We have changed to "delays APC/C dephosphorylation during mitotic exit". According to Fig 6D, the mutant APC/C (1-E562A) delays APC/C dephosphorylation because of stronger Plx1 binding which prevents efficient APC/C dephosphorylation during mitotic exit.

-Page 7: "anaphase extracts" are introduced without further clarifications, and this will be confusing, as most readers will associate anaphase with the period when Cyclin B is

degraded. The authors should clarify that these extracts are supplemented with non-degradable Cyclin B.

We added a description of anaphase extracts in the text as to “anaphase extracts supplemented with non-degradable cyclin B, to ensure continuation of the anaphase state even after activation of the APC/C ...”

-Probably the most striking effect described in this study arises from mutating Ser558 to Ala. This mutant ablated not only the interaction of PP2A-B56 with the Apc1-500 loop, but also the interaction of Plx1 with the upstream sites T532 and T539. The authors argue that Ser558, being part of a "degenerate" PBD binding site, LSP, may contribute to stabilization of Plx1 binding through its PBD. I note that the early work of Mike Yaffe and colleagues (Elia et al. 2003) seems to exclude this because any ligand with side chains other than Ser, even Thr, on the residue preceding the phospho-site were unable to interact with the PBD. Could it rather be that the LSPV sequence encompassing S558 is a docking site for the Cks2/Cyclin/CDK complex? A possible expectation if the authors hypothesis were correct is that T532 and T539 continue to be phosphorylated when Ser558 is mutated. A possible expectation of the alternative hypothesis I propose here is that these sites are not phosphorylated (even when PP2A activity is inhibited). Testing this should be within easy reach with the tools available to the authors.

First, we would like to point out our Fig EV4 data which show that S558A mutation has no impact on T532 and T539 phosphorylation: Both pT532 and pT539 phospho-site specific antibodies recognise the same levels of T532 and T539 phosphorylation as WT even when S558 is mutated.

Second, we have reconstituted PBD and Apc1-loop⁵⁰⁰ binding assay in the presence of Cdk2/cyclin A. In this assay, the effects of Cks, other kinases and phosphatases can be minimised. S558A mutation reduced its binding to nearly 50% even though both T532 and T539 sites (the PBD consensus sites) are well phosphorylated. The result is presented in new Appendix Fig S1 and Fig EV4B. In the same assay, we could confirm our result (Fig. 1C): 1) T539 phosphorylation is more important than T532 for Plx1 binding. 2) when both T532 and T539 are mutated (2A), Plx1 shows no binding to Apc1-loop⁵⁰⁰ even if S558 is phosphorylated. Thus, as appreciated in the field, the PBD consensus is vital for Plx1 binding. In addition, our result implies that phosphorylation of S558 plays a role for Plx1 binding towards the PBD-PBM interaction.

Third, although we appreciate the breakthrough discovery of PBD-phosphopeptide recognition by the Yaffe laboratory, Elia et al used short peptides (16 residues) in the binding assay. In contrast, we have used 69-residue fragments that might contain not only the genuine PBMs but also several possible Plx1 binding or recognition sites. Several recent reports suggest that a tandem orientation of Plk1-binding sites stabilises Plk1-binding (Jia et al 2015, Singh et al 2020), likely through PBD-PBD interaction (Zhu et al 2016).

At the moment, we do not know the underlying mechanism, but we speculate that the orientation of the sites (T539 and S558) and/or the distance between them may be just right for the formation of a hetero-trimer complex including Plx1 (two molecules) and Apc1-loop⁵⁰⁰. It is also possible that the requirement of Plx1-binding is slightly relaxed because of specific adjacent sequences or an as-yet unidentified mechanism. The detailed study of the

mechanism underlying this cooperative binding is intriguing, but we believe that this is beyond the scope of this manuscript.

-A related issue is raised from a statement in the Discussion: If two molecules of Plx1 bind to Apc1-loop 500 through T539 and S558, as proposed by the authors, why doesn't the T532A mutant, which cannot bind PP2A (like E562A) have at least as much Plx1 bound as WT?

In the reconstituted Plx1-PBD binding assay (Appendix Fig S1), as reviewer 3 expects, the T532A mutant can have as much Plx1 bound as WT when T539 and S558 are as well phosphorylated as WT (Fig EV4B). We surmise that slight reduction of Plx1 binding to the T532A mutant in anaphase extracts might be due to some reduction of pT539 phosphorylation (Fig EV4A), but we do not understand the exact mechanism.

-On page 9, the authors demonstrate that Plx1 can phosphorylate Apc3-loop, which is interesting, even if ultimately the authors do not test the significance of this observation. Regardless, a question for this and other kinase assays is how the authors think of the specificity of these phosphorylation events. We see Plx1 can phosphorylate targets, but is this selective for Plx1 or would other mitotic kinases, under similar conditions, do the same? Can the authors comment on this point?

In order to show the specificity of the in vitro phosphorylation by Plx1, alpha-casein, Cdc20 NTD (N159), MBP or Apc3-loop were incubated in the presence of ³²P-ATP and Plx1. With the exception of MBP, all substrates are phosphorylated by Plx1 in a time-dependent manner, in particular, Apc3-loop being the most phosphorylated among the four proteins tested. This result indicates that Plx1 phosphorylates its substrates with a certain specificity.

We think that different kinases have different substrate specificity, allowing different site phosphorylation. In addition, phosphorylation of some sites may require or prefer adjacent site phosphorylation by a primer kinase, which in turn attracts an additional kinase in order to specifically phosphorylate the site. For example, Apc3 can be phosphorylated by Cdk1 and Plx1, but we think that Cdk1- or Plx1-mediated phosphorylation sites are different and impacts on APC/C regulation might also be different. We believe that the detailed map and study of the impact of Plx1-dependent phosphorylation is beyond the scope of this manuscript.

-On page 10 and page 13 (two instances) the authors refer to an E532A mutation when I think they really mean E562A in all three cases. Please check.

Thank you very much for spotting this. We corrected them.

-Page 17: streamline sentence "In addition, another physically distant flexible loop from Apc1-loop, Apc3-loop, participates in the control...". Could rather read "In addition, Apc3-loop, another flexible loop physically distant from Apc1-loop, participates in the control..."

We corrected.

-Page 18: "Lue" should read "Leu"

We corrected.

-In general, many of the experiments here rely on the assumption that APC loop regions fused to MBP are reasonable proxies for their behavior on the APC/C. This is reasonable but the authors may elect to include a statement to warn readers that they are aware that this is not necessarily true.

In this revised manuscript, we have included more data using whole APC/C (Figs 2C, 5C and EV5 and Appendix Figs S2, S3, S4 and S8) and all the new results essentially support our model and the results obtained using APC/C loop regions fused to MBP. Yet, we concede that the reviewer raises a valid point, and thus we have added a statement in the Discussion.

Thank you for submitting your revised manuscript to The EMBO Journal. We have now heard back from the three original referees, and I am pleased to say that they all found the previously-raised points satisfactorily addressed. Following a final revision round to address some remaining minor issues noted mainly by reviewer 1, as well as the below-listed editorial points, we shall therefore be happy to accept the study for publication in our journal.

Referee #1:

The authors have done a really good job in dissecting the complexity of APC/C regulation with relation to Plx1 and PP2A-B56 binding to the APC1 loop. They have done a good job in addressing the reviewers points and I support publication. I have a few minor points:

- 1) The requirement for the LSPI sequence of the B56 motif for efficient Plx1 binding is intriguing. Could this relate to the discoveries from the Ventikaraman lab showing a binding pocket for hydrophobic residues in the PBD (Sharma et al 2019)?
- 2) I think there are some key experiments I requested that relates to the relevance of their discoveries in context of APC/C activity that has been hidden in EV and appendix figures - in my view these experiments are key and should be in main figures. This relates to all experiments testing APC/C activity in extract and ubiquitination assays. I must still point out that these are minor effects - one could consider to have a comment on this in the discussion. This does not rule out the importance of their discoveries and that they could be even more important in mitosis in somatic cells.
- 3) The fact that T532A does not bind B56 is interesting and the authors discuss different possibilities. Could an alternative explanation (although not very likely) be that the T532 phosphorylation engages the active site of PP2A to stabilise the interaction but is a poor substrate - we know from Arpp19 that this inhibits PP2A-B55 likely through a phosphorylation engaging the active site.
- 4) I would recommend minimising the amount of information in appendix figures and see if some of this can be moved to EV and the mains (see comment above).
[Editor's comment: we only allow 5 EV figures, so this may not be an option here]

Referee #2:

Overall the revised manuscript incorporates a fair amount of additional data where their findings are extrapolated to the full APC/C rather than the peptide. They have also adequately addressed my additional concerns. Therefore, I believe this study is suitable for publication.

Referee #3:

I am happy to support publication of this revised manuscript. The authors have addressed my concerns, either textually or through new experiments. I congratulate the authors for a technically excellent and insightful paper.

RE: Manuscript EMBOJ-2020-107516R
Point-by-Point response to reviewers' comments

Referee #1:

The authors have done a really good job in dissecting the complexity of APC/C regulation with relation to Plx1 and PP2A-B56 binding to the APC1 loop. They have done a good job in addressing the reviewers points and I support publication.

We thank the reviewer again for his constructive comments and suggestions which have greatly improved our work.

A have a few minor points:

1) The requirement for the LSPI sequence of the B56 motif for efficient Plx1 binding is intriguing. Could this relate to the discoveries from the Ventikaraman lab showing a binding pocket for hydrophobic residues in the PBD (Sharma et al 2019)?

Thank you for your suggestion. It is possible that the LSPI sequence of the B56 motif may be involved in efficient Plx1 binding via the hydrophobic pocket of the PBD. Yet, we do not have any experimental data to support or deny it at the moment. Further structural and biochemical analysis will be required for a better understanding of the detailed molecular mechanism.

2) I think there are some key experiments I requested that relates to the relevance of their discoveries in context of APC/C activity that has been hidden in EV and appendix figures - in my view these experiments are key and should be in main figures. This relates to all experiments testing APC/C activity in extract and ubiquitination assays. I must still point out that these are minor effects - one could consider to have a comment on this in the discussion. This does not rule out the importance of their discoveries and that they could be even more important in mitosis in somatic cells.

Thank you for your suggestion. Yet, main figures already have space constraints and thus there is no room for presenting the whole data, i.e. gel data such as APC/C activity assay and the quantification together. We think that it is better to present these data next to each other rather than a part of data squeezed into the mains. This can be achieved as EV or Appendix figures, allowing readers to smoothly follow the story. It is also true that EV and Appendix figures are easily accessible via the EMBO J website, in particular EV figures like the mains, so they are not hidden. However, the reviewer raises a valid point, and thus we have moved Appendix Fig S3B to EV2, EV1 to Appendix S1 and Appendix S5 to the main Fig 2E.

3) The fact that T532A does not bind B56 is interesting and the authors discuss different possibilities. Could an alternative explanation (although not very likely) be that the T532 phosphorylation engages the active site of PP2A to stabilise the interaction but is a poor substrate - we know from Arpp19 that this inhibits PP2A-B55 likely through a phosphorylation engaging the active site.

Thank you for your suggestion. We are also interested in how T532 phosphorylation promotes B56-binding and how phospho-T532 is dephosphorylated. Yet, at the moment we do not know the mechanism. Your alternative explanation is intriguing, but according to our results using a single B56 subunit, not a ternary holocomplex (Figure 7C), we surmise that B56 itself preferentially binds to T532-phosphorylated Apc1-loop⁵⁰⁰, independently of the active site of PP2A catalytic subunit. It would be a great idea to explore the detailed mechanism in future, including whether phospho-T532 can play a role like Arpp19 when B56 is in a ternary PP2A holocomplex.

4) I would recommend minimising the amount of information in appendix figures and see if some of this can be moved to EV and the mains (see comment above).

[Editor's comment: we only allow 5 EV figures, so this may not be an option here]

In order to address all reviewers' points, we have performed several new experiments and effectively located them in the main, EV or Appendix figures. We have tried to provide as much data as possible. However, we concede that the reviewer raises a valid point, and thus we have moved Appendix Fig S3B to EV2, EV1 to Appendix S1 and Appendix S5 to the main Fig 2E. As a consequence, we have managed to reduce the number of Appendix figures from nine to eight.

Referee #2:

Overall the revised manuscript incorporates a fair amount of additional data where their findings are extrapolated to the full APC/C rather than the peptide. They have also adequately addressed my additional concerns. Therefore, I believe this study is suitable for publication.

We thank the reviewer again for his/her constructive comments and suggestions which have greatly improved our work.

Referee #3:

I am happy to support publication of this revised manuscript. The authors have addressed my concerns, either textually or through new experiments. I congratulate the authors for a technically excellent and insightful paper.

We are very pleased to receive such positive comments. We thank the reviewer again for his/her constructive comments and suggestions which have greatly improved our work.

Thank you for submitting your final revised manuscript for our consideration. I am pleased to inform you that we have now accepted it for publication in The EMBO Journal.

Corresponding Author Name: Hiroyuki Yamano

Journal Submitted to: EMBO J.

Manuscript Number: EMBOJ-2020-107516